DOI: 10.1038/s41467-017-02168-x | OPEN

# Differentially evolved glucosyltransferases determine natural variation of rice flavone accumulation and UV-tolerance

Meng Peng[1], Raheel Shahzad[1], Ambreen Gul[1], Hizar Subthain[1], Shuangqian Shen[1], Long Lei[1], Zhigang Zheng[1], Junjie Zhou[1], Dandan Lu[1], Shouchuang Wang[1], Elsayed Nishawy[1,2], Xianqing Liu[1], Takayuki Tohge[3], Alisdair R. Fernie[4] & Jie Luo[1,5]

Decoration of phytochemicals contributes to the majority of metabolic diversity in nature, whereas how this process alters the biological functions of their precursor molecules remains to be investigated. Flavones, an important yet overlooked subclass of flavonoids, are most commonly conjugated with sugar moieties by UDP-dependent glycosyltransferases (UGTs). Here, we report that the natural variation of rice flavones is mainly determined by OsUGT706D1 (flavone 7-O-glucosyltransferase) and OsUGT707A2 (flavone 5-O-glucosyltransferase). UV-B exposure and transgenic evaluation demonstrate that their allelic variation contributes to UV-B tolerance in nature. Biochemical characterization of over 40 flavonoid UGTs reveals their differential evolution in angiosperms. These combined data provide biochemical insight and genetic regulation into flavone biosynthesis and additionally suggest that adoption of the positive alleles of these genes into breeding programs will likely represent a potential strategy aimed at producing stress-tolerant plants.

[1] National Key Laboratory of Crop Genetic Improvement and National Center of Plant Gene Research (Wuhan), Huazhong Agricultural University, Wuhan 430070, China. [2] Desert Research Center, Genetics Resource Department, Egyptian Deserts Gene Bank, Cairo 11735, Egypt. [3] Graduate School of Biological Sciences, Nara Institute of Science and Technology, Ikoma, Nara 630-0192, Japan. [4] Max-Planck-Institute of Molecular Plant Physiology, Am Mühlenberg 1, Potsdam-Golm 14476, Germany. [5] Institute of Tropical Agriculture and Forestry, Hainan University, Haikou, Hainan 572208, China. Meng Peng, Raheel Shahzad, Ambreen Gul and Hizar Subthain contributed equally to this work. Correspondence and requests for materials should be addressed to J.L. (email: jie.luo@mail.hzau.edu.cn)

Plants as sessile species produce hundreds of thousands of different phytochemicals, especially secondary or specialized metabolites, which play vital roles in the adaptation of plants to the changing environments[1,2]. Although we know that the majority of their chemodiversity comes from decoration of a much smaller number of core skeleton precursor molecules, our understanding of the genetic factors that shape this chemodiversity remains limited[3]. Flavonoids are a group of plant secondary metabolites playing important roles in a wide range of physiological processes[4–6] including ultraviolet-B (UV-B, 280–315 nm) protection[7,8] which is becoming increasingly important for yield stability in the face of present levels of environmental deterioration[9,10]. In addition, their beneficial effects on human health as a dietary source of antioxidants attract increasing attention[11,12]. Flavonoids consist of a common C6–C3–C6 structure with two benzene rings (A ring and B ring) interconnected by a three-carbon heterocyclic pyran ring (C ring) and can be subdivided into six major groups, namely flavones, flavanones, flavonols, flavanols, anthocyanins, and isoflavones according to structural considerations (Supplementary Fig. 1).

Widely distributed in the plant kingdom, flavonoids usually exist in their decorated forms catalyzed by glycosyltransferases, acyltransferases, hydroxylases, and methyltransferases[13,14]. Glycosylation of flavonoids confers structural complexity, as well as molecular solubility and stability, subcellular transportability, and biological activity. For example, glycosylation of anthocyanins has been demonstrated to play pivotal roles in both the solubility and stability of these pigments in a range of flowers[15]. Zhao et al. reported that multidrug and toxin extrusion transporter (MATE) 1 and 2 show higher transport capacity for epicatechin 3′-glycoside and anthocyanin glycosides, respectively[16,17]. Furthermore, the bitterness in Citrus species is caused by regiospecific flavanone glycosylation[18], while flavonol 3-O-glycosides have been demonstrated to impart the astringent taste sensation in infusions of black tea[19].

Glycosylation of flavonoids is usually catalyzed by UDP-dependent glycosyltransferases (UGTs)[20] although exceptions have also been reported in carnation (Dianthus caryophyllus)[21], rice (Oryza sativa)[22], and Arabidopsis (Arabidopsis thaliana)[23]. UGTs typically transferring sugars onto small-molecule acceptors including plant secondary metabolites belong to Family 1 of the glycosyltransferases super-family (CAZy database; http://www.cazy.org/GlycosylTransferases.html). They are largely characterized by a C-terminus signature motif compromising 44 amino acid residues (Plant Secondary Product Glycosyltransferase (PSPG) box)[24]. A number of flavonoid UGTs have been characterized in a wide range of plant species, including crops, ornamental and medical plants[14]. These UGTs include flavonoid 3-O-glycosyltransferases (3GTs), flavonoid 5-O-glycosyltransferases (5GTs), flavonoid 7-O-glycosyltransferases (7GTs), flavonoid 3′-O-glycosyltransferases (3′GTs), flavonoid glycoside glycosyltransferases (GGTs) and flavonoid C-glycosyltransferases (CGTs). Phylogenetic analyses of all characterized flavonoid UGTs have suggested that they can be divided into group of enzymes with different sugar-attachment specificities[25,26], However, it worth noting that this scenario is based on results exclusively from UGTs with flavonol and/or anthocyanin substrate preferences.

To date, knowledge on flavonoid UGTs comes almost exclusively from researches on flavonols which accumulate to reasonable abundance in Arabidopsis[27–31], while the biochemical aspects of glycosylated flavones, ubiquitously distributed in grass crops but almost absent in Brassicaceae[32,33], remain largely unknown. For example, few flavone-active UGTs have been characterized and the natural genetic variation of rice flavones and the in planta biological function of glycosylated flavones are largely unknown.

We describe here the genetic and biochemical dissection of natural variation in the major rice flavones and the characterization of UGTs in a new clade of flavonoid glucosyltranferases including OsUGT706D1 (F7GlcT) and OsUGT707A2 (F5GlcT). We show that F7GlcT and F5GlcT enzymes evolved independently in plants, and furthermore, provide evidence that functional allelic variation in these genes corresponds to the level of UV exposure and that lines overexpressing OsUGT706D1 and OsUGT707A2 have enhanced UV tolerance, indicating that tapping the natural variation inherent in these genes may represent a useful route to provide food security.

## Results

**Flavonoid profiling among species.** Application of a widely-targeted liquid chromatography-mass spectrometry (LC-MS) method[34] resulted in the comprehensive profiling of flavonoid in 14 plant species spanning the pre-angiosperm, monocot and eudicot clades, namely fern (Nephrolepis auriculata), peat moss (Sphagnum palustre), citrus (Citrus sinensis), populus (Populus deltoides), potato (Solanum tuberosum), tobacco (Nicotiana tabacum), Arabidopsis, palm (Petiolus trachycarpi), maize (Zea mays), bamboo (Phyllostachys pubescens), wheat (Triticum aestivum), hulless barley (Hordeum vulgare), rice and Brachypodium (Brachypodium distachyon). A total of 85 flavonoids belonging to the three subclasses—flavone, flavonol, and flavanone—were detected in our study (Supplementary Data 1) and their profiles were visualized by hierarchical cluster analysis (HCA, Fig. 1a). On the basis of their plant-specific accumulation patterns, flavonoids could be clearly grouped into six subclusters, reflecting three common modifications (Fig. 1a). Most flavonoids could be glycosylated and their glycosides were grouped into distinct clades depends on the core flavonoid skeletons. Flavonoids in cluster 1 accumulate most prominently in monocot plants and are mainly represented by tricin O-glycosides, including tricin 5-O-hexoside, tricin 5-O-hexosyl-O-hexoside, tricin 7-O-hexoside and tricin 7-O-rutinoside (Supplementary Data 2). Flavonoids in cluster 2 are mainly represented by flavone O-glycosides, such as apigenin 7-O-glucoside, apigenin 5-O-glucoside, luteolin 7-O-glucoside, chrysoeriol 7-O-hexoside, with the highest levels detected in monocots followed by citrus, populus and peat moss. By contrast, flavonoids in cluster 3 are mainly composed of flavonol-O-glycosides, including quercetin 7-O-hexoside, quercetin 3-O-glucoside-7-O-glucoside and quercetin 5-O-hexoside and were commonly distributed in all species studied here. Flavonoid O-glycosides could be further modified by acylation and acylated flavonoid O-glycosides show distinct accumulation patterns (clusters 4 and 5, Supplementary Data 2). Most flavonoids in cluster 4 are malonyl flavonoid glycosides, which accumulated in citrus, populus, and peat moss, whereas hydroxycinnamoyl flavonoid glycosides, as representative flavonoids of cluster 5, are cereal-specific. Additionally, flavonoids in cluster 6 are generally tricin-lignans which accumulated to high levels in monocots (Supplementary Data 2), where they have been demonstrated to be involved in lignification[35]. Despite their low-level accumulation, we managed to detect five flavones in Arabidopsis, including two chrysoeriol derivatives, two tricin-glycosides and one tricin-lignan (Supplementary Fig. 2). We subsequently choose rice, one of the main model monocots, which has the highest accumulation of most flavones, and is of massive importance to the human diet, for assessing the natural variation in flavone content and dissecting its genetic and biochemical bases.

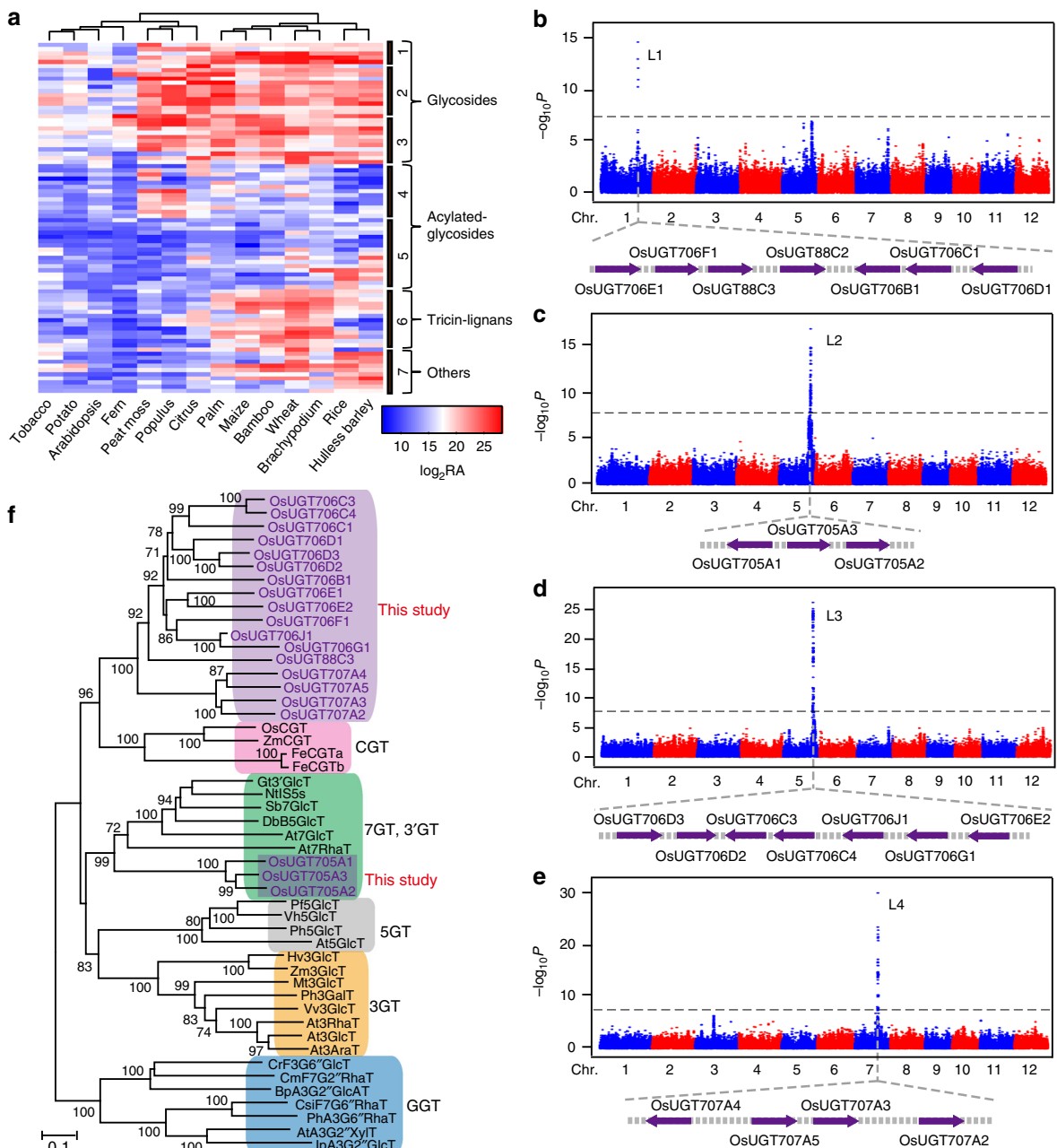

**Fig. 1** Identification and phylogenetic analysis of flavonoid UGTs in rice. **a** LC-MS profiling of flavonoids in 14 species across the plant kingdom. Heat map visualization of relative differences of flavonoids by the average of all values in two biological replicates. Scatterplots of association results for 4 flavonoid traits across the 12 rice chromosomes. The strength of association is indicated as the negative logarithm of the P value for the linear mixed model. All metabolite-SNP associations with P values below $6.6 \times 10^{-8}$ (horizontal dashed lines in all Manhattan plots) are plotted against genome location in intervals of 1 Mb. Traits are as follows: apigenin 7-O-glucoside in leaf (**b**), tricin O-hexoside-O-hexoside in seed (**c**), chrysoeriol O-feruloylhexoside-O-hexoside in leaf (**d**), apigenin 5-O-glucoside in leaf (**e**). The Manhattan plots from six individual replicate (two replicates in leaf and four replicates in seed) for each locus are provided in Supplementary Fig. 3. Candidate glycosyltransferases within each locus are in purple. **f** An unrooted phylogenetic tree was constructed as described in Methods. OsUGT88C2, which shows 89% identities to OsUGT88C3, was not shown in this tree as its short protein length. Bootstrap values >70% (based on 1000 replications) are indicated at each node (bar: 0.1 amino acid substitutions per site). RA, relative peak area

**Genetic variation of rice flavonoids**. To investigate the genetic control of natural variation in the major flavonoids in rice, the metabolite profiling data and high-quality SNPs with minor allele frequency (MAF) ≥5% were obtained from previous studies using a worldwide collection of 529 rice accessions[36,37] and a metabolite-based genome-wide association study (mGWAS) was performed by a gene-based analysis[38]. The association results showed that natural variation in rice flavonoids was mainly controlled by four prominent loci (Fig. 1b–e; Supplementary

Data 3). In searching for candidate genes underlying these loci, we found tandem annotated UDP-dependent glycosyltransferase encoding genes within the confidence intervals of each locus (Fig. 1b–e). For example, four UGT-encoding genes locate on chromosome 7 (locus 4) are significantly associated with the levels of various flavone 5-O-hexoses such as apigenin 5-O-glucoside, chrysoeriol 5-O-hexoside, and tricin 5-O-hexoside (Fig. 1e, Supplementary Fig. 3a, b and Supplementary Data 3). Therefore, we posit that these four UGTs are putative flavone

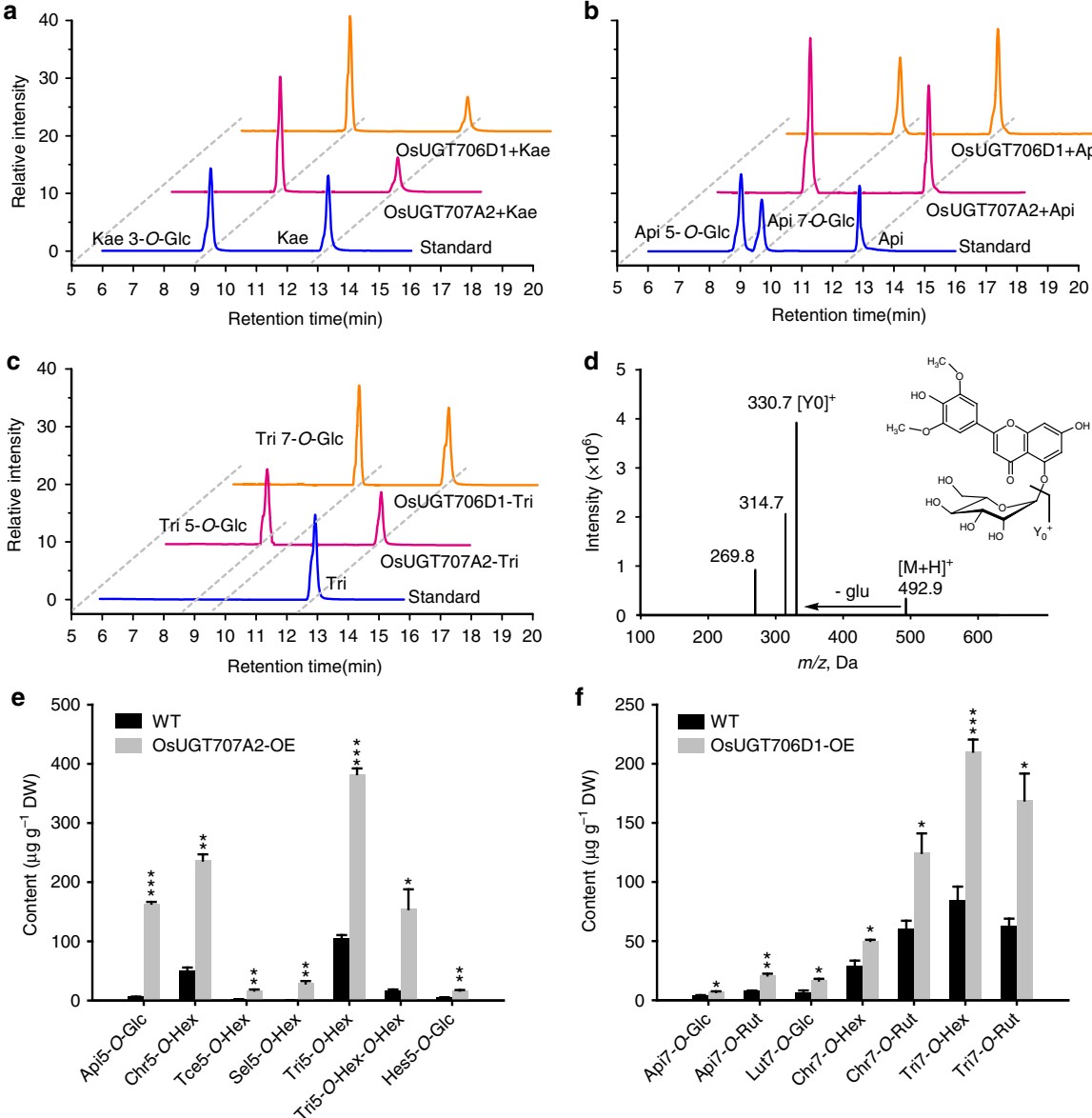

**Fig. 2** Functional analysis of *OsUGT706D1* and *OsUGT707A2*. HPLC chromatograms of the reaction of OsUGT706D1 and OsUGT707A2 with UDP-glucose and different substrates, kaempferol (**a**), apigenin (**b**) and tricin (**c**). Chromatograms show the absorption at 330 nm. **d** The MS spectrum and chemical structure of the product in the reaction of OsUGT707A2 with UDP-glucose and tricin. Bar plots for the contents of flavone 7-*O*-glycosides and flavonoid 5-*O*-glycosides in overexpression lines of OsUGT707A2 (**e**) and OsUGT706D1 (**f**). The data are presented as mean ± SD, $n = 3$. *$P < 0.05$, **$P < 0.01$, ***$P < 0.001$, Student's *t* tests. *Api* apigenin; *Kae* kaempferol; *Tri* tricin; *Sel* selgin; *Tce* 3′,4′,5′-tricetin; *Glc* glucoside; *Hex* hexoside; *Rut* rutinoside, *Chr* chrysoeriol; *Hes* hesperetin; *WT* the transgenic background variety ZH11

5-*O*-glucosyltransferases underlying this locus. We also found strong association between flavone 7-*O*-glucosides and *OsUGT88C3* (*Os01g53370*) for locus 1 (Fig. 1b, Supplementary Fig. 3a, b and Supplementary Data 3). Similarly, significant association between seven UGTs encoding genes in locus 3 and the amount of tricin *O*-hexoside-*O*-glucoside (Student's *t* tests, $P < 1.53 \times 10^{-15}$; Fig. 1d, Supplementary Fig. 3b and Supplementary Data 3) and the association between three tandem UGT-encoding genes and the levels of chrysoeriol *O*-feruloylhexosyl-*O*-hexoside in locus 2 suggest these UGTs are candidates for the corresponding GWAS hits (Fig. 1c, Supplementary Fig. 3a and Supplementary Data 3). In total, 21 putative UGT-encoding genes were assigned to be the candidates underlying four identified loci located on chromosomes 1, 5 and 7.

**Characterization of putative flavonoid UGTs.** Previous phylogenetic analyses have categorized flavonoid UGTs into unique clusters based on their regiospecificity (i.e., the position of glycosylation) for sugar acceptors, including 3GT, 5GT, 7GT, 3′GT and GGT subfamilies[14]. To figure out which group our putative UGTs belong to, we generated a phylogenetic tree by neighbor-joining algorithm using amino acid sequences of the 29 previously reported flavonoid UGTs and the candidates identified in this study (Fig. 1f). We found that three of the putative UGTs, OsUGT705A1, OsUGT705A2, and OsUGT705A3, grouped into the 7GT/3′GT clade mainly formed by previously characterized flavonoid 7-*O*-glycosyltransferases and flavonoid 3′-*O*-glycosyltransferases (Fig. 1f). However, the remaining 18 putative UGTs formed a distinct clade comprising of no characterized UGTs

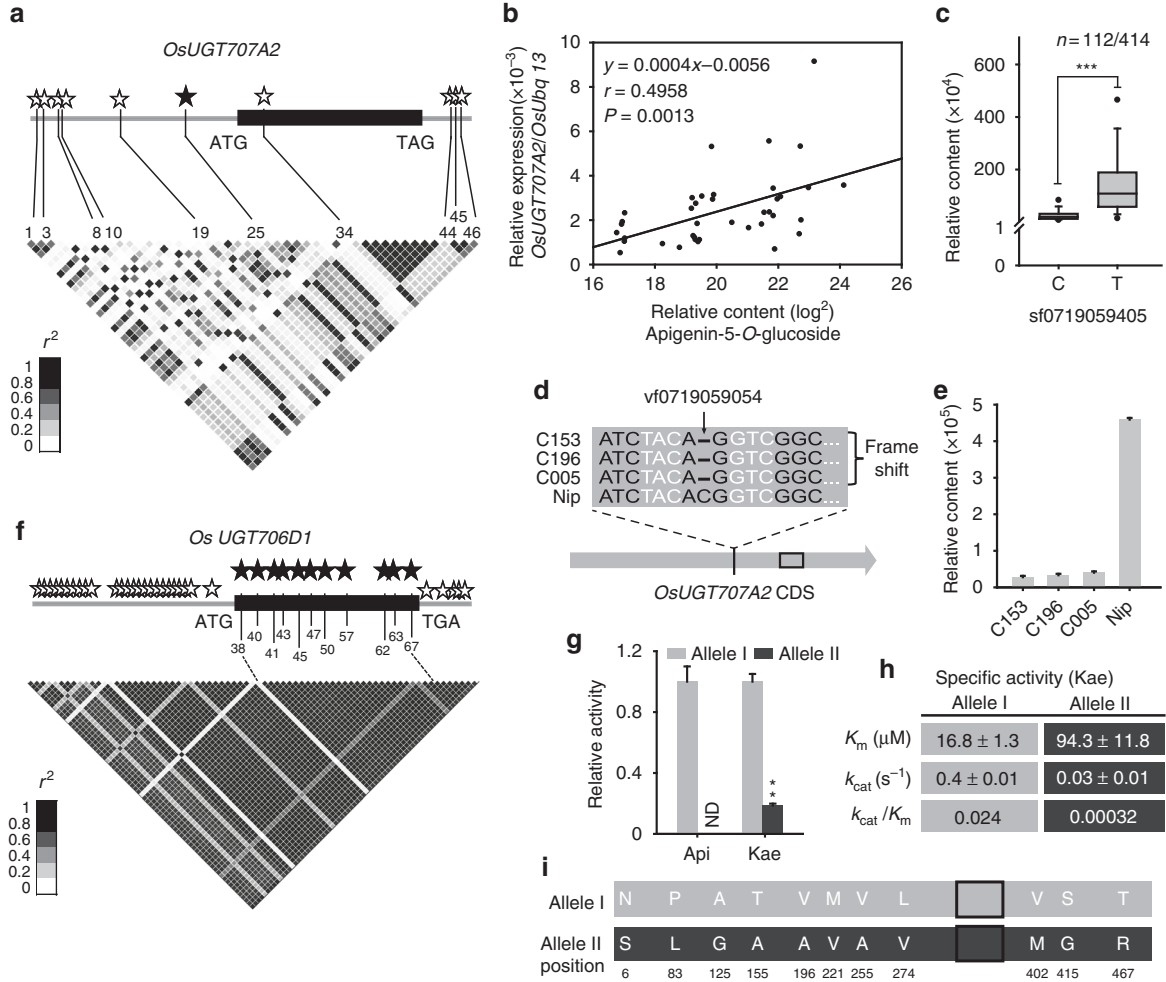

**Fig. 3** Analyses of functional SNPs within *OsUGT707A2* and *OsUGT706D1*. A representation of the pairwise $r^2$ value (a measure of linkage disequilibrium) among polymorphic sites in *OsUGT707A2* (**a**) and *OsUGT706D1* (**f**), where the darkness of the box corresponds to the $r^2$ value according to the legend. Stars represent the significant sites identified in *UGT707A2* ($P < 8 \times 10^{-33}$) and *UGT706D1* ($P < 2 \times 10^{-19}$), and solid stars indicate the functional SNPs. The $P$ value is calculated using the Student's $t$ tests. **b** Correlation between relative contents of apigenin 5-*O*-glucose and transcript levels of OsUGT707A2 in 39 rice varieties. **c** Box plot for the content of apigenin 5-*O*-glucoside; plotted as an associated site at Chromosome 7. sf0719059405. \*\*\*$P < 0.001$, Student's $t$ tests. **d** Alignment of partial nucleotide sequences of *OsUGT707A2* from different rice varieties. C153, C196 and C005 indicate three rice varieties that originated in china, respectively. **e** Comparison of the content of apigenin 5-*O*-glucoside in different rice varieties. The data are presented as mean ± SD, $n = 3$. **g** Enzymatic activity of OsUGT706D1 from two alleles. Allele I indicates the high producer genotype whereas the allele II indicates the low producer genotype. Assay was repeated three times and bars indicate mean ± SD, $n = 3$. \*\*$P < 0.01$, Student's $t$ tests. **h** Kinetic parameters of OsUGT706D1 from two alleles. The data are presented as mean ± SD from two replicate independent assays. **i** Alignment of full amino acid sequences of OsUGT706D1 from two alleles. Black block indicates the conserved PSPG box domain. *Nip* nipponbare, *ND* not detected

(Supplementary Fig. 4), suggesting that it is highly likely that these putative flavonoid UGTs display novel specificities.

In order to functionally characterize these putative UGTs, their open reading frames were successfully amplified and cloned into expression vectors with glutathione *S*-transferase (GST) fused to the *N*-terminus. In all, 20 recombinant UGTs that were successfully expressed in *E. coil* were purified for enzymatic assays. Two sugar donors (UDP-glucose and UDP-galactose) and six typical flavonoid aglycones, including flavones (apigenin and luteolin), flavonols (kaempferol and quercetin), and flavanones (naringenin and eriodictyol) were tested as potential substrates. In term of sugar acceptor, nine of these UGTs exhibit specific activity on flavonol aglycones. Among them, OsUGT706D2 showed the highest catalytic efficiency exhibiting the highest $k_{cat}/K_m$ value, followed by OsUGT706B1, OsUGT707A3 and OsUGT706C3 (Supplementary Table 1). Both OsUGT706C1 and OsUGT706A5 exhibited low activities on kaempferol, whereas OsUGT706F1 could utilize both kaempferol and quercetin as substrates

(Supplementary Fig. 5 and Supplementary Table 1). Five UGTs were identified to be active on both flavone (apigenin) and flavonols. OsUGT706E1 displayed higher activities using kaempferol than apigenin as substrate, although its $k_{cat}/K_m$ value for quercetin was with 10-fold lower (Supplementary Table 2). Similar results were obtained for OsUGT705A2 and OsUGT706C4 (Supplementary Table 2). By contrast, recombinant OsUGT706D1 and OsUGT707A2 exhibited higher activities with flavones than flavonols as substrates albeit with different regio-selectivity. For instance, both OsUGT706D1 and OsUGT707A2 reaction products gave a single peak that had the same retention time (RT; 9.4 min) and UV-spectra as kaempferol 3-*O*-glucoside when using kaempferol as a substrate (Fig. 2a). However, OsUGT706D1 reaction product generated a peak different from that of OsUGT707A2 with flavone (apigenin) as substrate. The former product eluted at 9.3 min had the same RT, fragmentation pattern and UV-spectra with an authentic apigenin 7-*O*-glucoside standard. The latter, however, eluted earlier than apigenin 7-*O*-glucoside and was

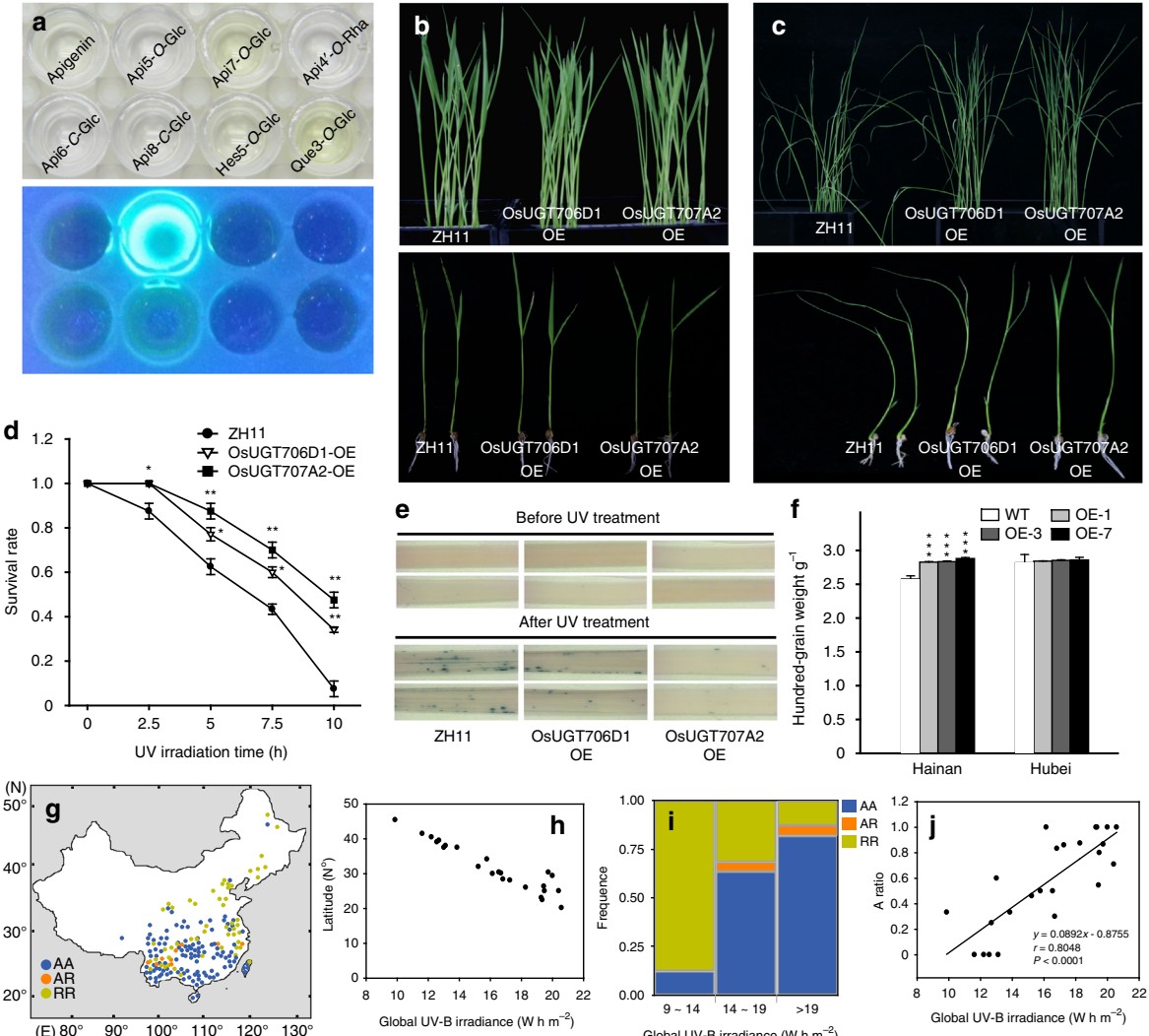

**Fig. 4** Functional characterization of the role of flavone glycosides and *OsUGT706D1* and *OsUGT707A2* in protection against UV-B irradiation in rice. **a** Flavonoid aglycone and glycosides under white light (upper) and UV-B (302 nm, lower). Concentration of flavonoids in each well was 2 μM. Phenotypes of rice seedlings (4-week old) before UV-B treatment (10 h, 20 W m$^{-2}$) (**b**) and after UV-B treatment (**c**). **d** Effect of UV-B irradiation on growth of rice seedlings. Experiment was repeated three times with 20 individuals in each group. Bars indicate means ± SD ($n = 3$). *$P < 0.05$, **$P < 0.01$, Student's $t$ tests. **e** NBT staining of rice seedlings before and after UV-B treatment. A 3–4 cm portion of the longest leaf was captured for comparison. **f** Plots for hundred-grain weight of *OsUGT707A2* overexpression plants and WT. Seeds were collected and measured on three individual plants per line. The data are presented as mean ± SD ($n = 3$). ***$P < 0.001$, Student's $t$ tests. **g** Geographical distribution of the two-gene combinations in 177 rice varieties in China. The different colors represent different combinations of *OsUGT707A2* and *OsUGT706D1*. R is the reference allele, and A is the alternate allele. **h** Plot for relationship between the average of global UV-B irradiance measured in 1990 (Soda (http://www.soda-is.com)) and latitude in China. **i** Allele distribution of varieties containing R or A allele associated with the average of daily UV-B irradiance measured at 1990. **j** Correlation for A ratio and the average of daily UV-B irradiance. The $r$ value is based on the Pearson correlation coefficient. The $P$ value is calculated using the Student's $t$ tests

identified as apigenin 5-*O*-glucoside using an authentic standard (Fig. 2b). These observations indicated that OsUGT706D1 catalyzed the glycosylation of the 7-OH group of flavone, whereas OsUGT707A2 transfers a glucose group onto the 5-OH group of flavone, which is consistent with the prediction from the association mapping. To investigate this further, another *O*-methylflavone, tricin, was tested. Each reaction gave a single peak with different retention time (Fig. 2c). LC-MS analysis showed that the OsUGT707A2 reaction product gave rise to a molecular ion at mass to charge ratio of 492.9[M + H]$^+$, which was consistent with the mass calculation of tricin glucoside (C$_{23}$H$_{24}$O$_{12}$, Fig. 2d). The MS$^2$ spectra displayed an identical fragmentation profile to authentic tricin 5-*O*-glucoside, in which the Y$_0^+$ ion was observed at *m/z* 330.7 due to the losses of the glucose moieties from *O*-glycosylation of the phenolic hydroxyl (−162, Fig. 2d). These

observations indicate that OsUGT706D1 and OsUGT707A2 are the two major flavone UGTs responsible for glucosylation 7-OH and 5-OH groups of rice flavones, respectively. No activities were detected for the remaining UGTs with any of the substrates tested. With regard to the sugar donor, catalytic activities were only observed with UDP-glucose but not with UDP-galactose.

Given that flavones constitute the majority flavonoids in rice (Fig. 1a), we next investigated the activities of flavone UGTs in planta by overexpressing them independently in ZH11, a widely used host cultivar for rice transformation. Flavonoid-targeted profiling was subsequently performed by LC-MS to determine the differences in flavonoid profile between the wild type and overexpression lines (Supplementary Fig. 6). Despite being able to utilize both flavones and flavonols in vitro, only flavone over-accumulation was observed in the UGT transgenic lines. Levels of

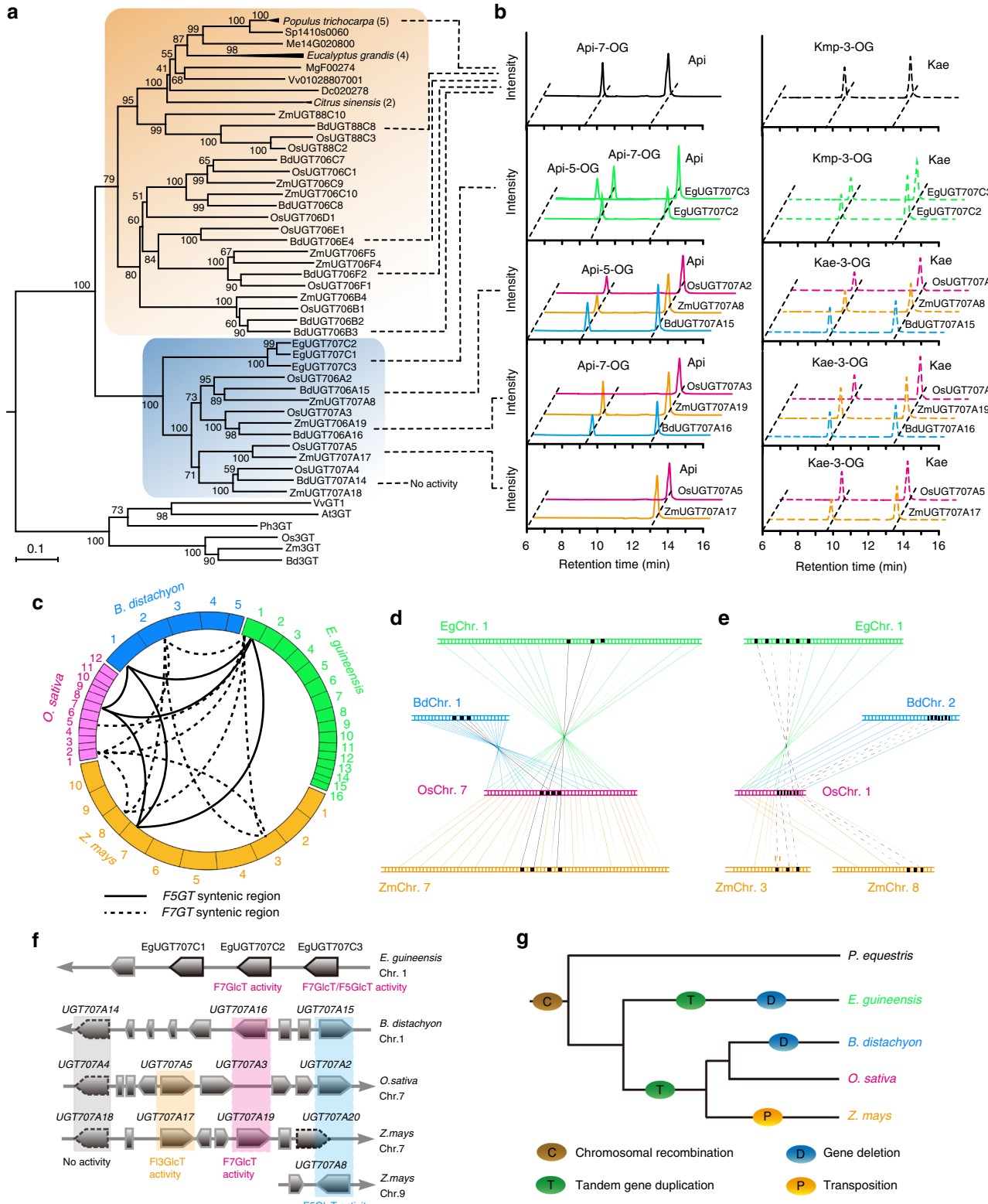

**Fig. 5** Genomic structure and function analysis of the *F7GlcT* and *F5GlcT* gene cluster region in plants. **a** Phylogenetic analysis using amino acid sequence of UGT genes obtained from different plant species. Scale implies amino acid substitutions per site. **b** In vitro biochemical assays of proteins from populus, palm, maize and Brachypodium. **c** Gene syntenic regions found in plant species. Solid line indicates *F5GlcT* syntenic regions and dashed line indicates *F7GlcT* syntenic regions. Syntenic gene linkage between plant species for *F5GlcT* (**d**) and *F7GlcT* (**e**). **f** Graphical map of *F5GlcT* cluster found among Commelinaceae species. *EgUGT707C1* was not successfully cloned. **g** Hypothetical scheme of evolutionary events in *F5GlcT* gene cluster region among monocot plants

flavone 5-O-glucosides were dramatically increased in OsUGT707A2 overexpression lines, suggesting OsUGT707A2 act as a flavone 5-O-glucosyltransferase in vivo (Fig. 2e). Similarly, metabolic profiling revealed an increased accumulation of flavone 7-O-glucosides and flavone 7-O-rutinoside in OsUGT706D1-overexpression plants (Fig. 2f), mirroring its in vitro activity. Analyzing the genes at the level of expression, we found OsUGT707A2 and OsUGT706D1 were more highly expressed than the other duplicated UGTs (Supplementary Fig. 7). Taken together, these results allowed us to conclude that OsUGT707A2 (termed flavone 5-O-glucosyltransferase, F5GlcT) and OsUGT706D1 (termed flavone 7-O-glucosyltransferase, F7GlcT) are the major enzymes controlling the natural variation of flavones in rice.

**Polymorphisms underlying variation of major rice flavonoids**. To investigate functional variations of OsUGT707A2 and OsUGT706D1, we searched for possible function polymorphism (s) within these two genes, including their promoter regions and UTRs using the RiceVarMap database[39]. We discovered 47 SNPs and 9 InDels (MAF ≥ 5%) in OsUGT707A2 (Supplementary Data 4), and 10 SNPs were significantly associated with the content of apigenin 5-O-glucoside (Student's $t$ tests, $P < 8 \times 10^{-33}$; Fig. 3a). We next tested whether the variation of their expression (SNPs in promoters) and/or coding variation contribute to the differences in rice flavonoid levels. For OsUGT707A2, we first measured the transcription levels of OsUGT707A2, as quantified by real-time quantitative reverse transcription PCR (qRT-PCR) across 39 rice varieties. As shown in Fig. 3b, correlation tests revealed that the expression level of OsUGT707A2 is significantly (Student's $t$ tests, $P < 0.01$) correlated with the content of apigenin 5-O-glucoside. Further investigation revealed that the SNP, sf0719059405, 400 bp upstream of OsUGT707A2 coding region, was strongly associated with the content of apigenin 5-O-glucoside (Fig. 3c) and therefore could represent the functional polymorphisms underlying this locus. However, even though strong associations between eight SNPs in the coding region and the level of apigenin 5-O-glucoside were observed (Supplementary Data 4), when the enzymatic activities of the two allelic recombinant proteins were compared, similar kinetic parameters were obtained (Supplementary Table 3). In addition, we found an low-frequency InDel (MAF = 0.629%, confirmed by re-sequence) in the coding sequence of UGT707A2, vf0719060607 (a C deletion, corresponding to position 773 bp), caused a frameshift and resulted in a truncated protein lacking the conserved UDP-glucose binding domain (Fig. 3d) in the three varieties, which display the lowest levels of apigenin 5-O-glucose (Fig. 3e). Taken together, results from qRT-PCR, kinetics and sequencing studies suggest that the SNP, sf0719059405, and InDel, vf0719060607, could be the functional polymorphisms responsible for the variation of apigenin 5-O-glucoside levels in rice. However, the effects of unidentified polymorphisms cannot be ruled out, nor can the possible roles of the "silent" ones which may affect RNA levels or splicing.

Similar analysis was also performed for OsUGT706D1. We found 74 SNPs and 7 InDels (MAF ≥ 5%) in OsUGT706D1 (Supplementary Data 5) and 60 SNPs were significantly associated with the content of apigenin 7-O-glucoside (Student's $t$ tests, $P < 2 \times 10^{-19}$; Fig. 3f). We observed no significant correlation between the transcript level of OsUGT706D1 and the accumulation of apigenin 7-O-glucoside (Supplementary Fig. 8). However, when the two alleles of OsUGT706D1 were cloned and expressed, the resultant recombinant proteins exhibited distinct catalyst efficiency, with a massive 100 fold difference in their $k_{cat}/K_m$ values (Fig. 3g, h). We identified 14 non-synonymous SNPs in the coding sequence, 11 of which were

significant associated with the content of apigenin 7-O-glucoside (Student's $t$ tests, $P < 10^{-19}$, Fig. 3i and Supplementary Data 5), implying that they could be functional polymorphisms responsible for natural variation of flavone 7-O-glucoside (apigenin 7-O-glucoside). Protein modeling showed that 14 of these amino acid residues are distant from the active site, and the remaining one, Leu83Pro, is located in the putative acceptor binding pocket that recognizes flavonoid (Supplementary Fig. 9). Similar results suggesting the impact of distal mutations on protein catalytic properties have been reported previously[40,41].

**Flavone O-glucoside content correlates with UV-B tolerance**. Although flavonols have long been reported to involve in UV-protection in many plant species by absorbing UV light in the 280–350 nm wavelength, it remains unclear whether flavones confer UV resistance in plants[32,42]. To this end, we measured the absorption of UV-VIS spectra by apigenin aglycone and its glycosides (Supplementary Fig. 10a). We found all of them gave similar λmax as those of flavonols[7,43]. Interestingly, however, we observed that apigenin 5-O-glucoside emitted a bright yellow fluorescence under UV-B light (302 nm) whereas the other flavones do not (Fig. 4a). With the purpose of testing the UV-protective function of flavone O-glucosides experimentally, we grew the overexpression lines of OsUGT707A2 and OsUGT706D1 alongside the wild-type plants under UV-B light. As shown in Fig. 4b, all plants grew normally and no significant difference was observed between the transgenic and the wild-type lines before UV treatment. However, most of the OsUGT707A2 and OsUGT706D1 overexpression plants survived with green leaves, while the wild-type plants became dramatically withered after UV-B irradiation although a similar amount of total flavonoids was detected in the transgenic and WT plants (Fig. 4c and Supplementary Fig. 10b). In addition, OsUGT707A2 overexpression lines exhibited significantly higher survival rate than that of OsUGT706D1 overexpression plants (Fig. 4d), suggesting that plants overaccumulating flavone 5-O-glucosides displayed enhanced UV-B tolerance of treatment than that of the plants which accumulated only flavone 7-O-glucosides. We then performed nitroblue tetrazolium (NBT) staining to detect superoxide anion radicals in the leaves of these plants and found significantly more blue precipitate in the wild-type seedlings than their transgenic counterparts and that the OsUGT707A2 overexpressing plants produced less precipitate than OsUGT706D1 (Fig. 4e). Furthermore, when WT and OsUGT707A2 overexpressing lines were grown in Hainan Island (20.2°N, 110.3°E) and Wuhan (30.3°N, 114.2°E), OsUGT707A2 overexpression lines showed significantly higher hundred-grain weight than those of WT plants in Hainan Island (which receives relatively high levels of UV-B irradiance), while no significant difference was observed for plants grown in Wuhan (which receives relatively low levels of UV-B irradiance) (Fig. 4f). Taken together, these results suggested that O-glycosylated flavones play a positive role in plant UV-B protection and flavone 5-O-glucosides are more effective UV-B protectants than flavone 7-O-glucosides. These results are interesting in light of a recent study in Arabidopsis which revealed that phenylacylation of flavonols catalyzed by a recently evolved phenylacyltransferase also rendered them as more effective UV-B protectants[44].

To see whether role of flavone glycosides play similar role in natural conditions, the high and low producers, based on the genotypes of selected SNP combinations within OsUGT707A2 (sf0719059405) and OsUGT706D1 (sf0130711475), were marked on a map of China according to their site of origin (Fig. 4g). In general, the different combinations were found to be associated with the pattern of UV irradiation. Most of the accessions

deriving from high irradiation areas such as south china (latitude 20°~25°) carry the AA combination (higher flavone glucoside levels) (Fig. 4g, h). By contrast, the RR combination (less flavone glucosides) accessions were mainly present in low irradiation habitats such as the north of China (latitude 35°–46°) (Fig. 4g, h). This is consistent with the significant correlation value between UV-B intensity and A/R ratio (Fig. 4i, j). The similar trend was also observed when a larger population of 345 cultivated accessions was examined in Europe and Asia (Supplementary Fig. 10c, d). Taken together, the abundance of the flavone O-glucosides appears to be related to adaptation to UV-B irradiance. However, we should stress at this point that many of the results reported here were from greenhouse grown plants and recent studies[9,45] have shown that translation of such results into a field environment is often not absolute.

**Comparative genomic signatures in UGT regions**. To better understand the evolutionary context of the UGTs at loci 1, 3 and 4 (F5GlcT and F7GlcT), which cluster distinctively from other reported UGTs (Fig. 1f and Supplementary Fig. 4), we performed an extensive BLAST search for homologies in Phytozome and NCBI databases using OsUGT707A2 and OsUGT706D1 as baits, respectively. For F5GlcT, we found at least three candidates (also in clusters) exist in oil palm (Elaeis guineensis) and eight in Gramineae plants. By contrast, numerous F7GlcT homologs were obtained among both monocots and eudicots such as rice, maize, oil palm, orange, populus, and grape (Vitis vinifera). We subsequently performed a protein phylogeny reconstruction based on a collection of these homologs and additional flavonoid 3-O-glycosyltransferases (F3GTs) sequences from Arabidopsis, grape and monocots to improve the strength of the sequence clustering (Fig. 5a). The F7GlcT clade was formed by widely-distributed F7GlcT proteins found in monocots and eudicots (Fig. 5a) and at least 5 UGTs were biochemically proven to show F7GlcT activities (Fig. 5b), suggesting that F7GlcT function is conserved in the Angiosperm. By contrast, F5GlcT genes formed a strongly clustered clade unique to the Commelinids and independent to the F3GTs and F7GlcTs (Fig. 5a), confirming the exclusive identification of F5GlcT orthologs. Furthermore, this similarity based tree reflect s an APGIII taxonomic relationship, suggesting early appearance of F5GlcT in a common monocot ancestor, in accordance with the apparent absence of F5GlcT in Gymnosperms and core Eudicotyledons. However, the LC-MS profile of leaves of Chinese cymbidium (Cymbidium sinense) did not allow identification of flavone 5-O-glucosides (Supplementary Fig. 11), suggesting that F5GlcT evolved after speciation between Orchidaceae and Commelinaceae. Additionally, three tandem UGTs harbored in Elaeis guineensis belong to a single clade, separated from those found in the Gramineae (Fig. 5a). These observations suggest F5GlcT genes evolved independently after speciation between Palmaceae and Gramineae and are not duplicated in the same manner in Commelinids. To confirm this hypothesis, we took advantage of the available genome data and performed co-linear genomic analyses among Elaeis guineensis, Oryza sativa, Zea mays and Brachypodium distachyon (Fig. 5c). F5GlcT syntenic blocks were found on OsChr7, ZmChr7, BdChr2 and EgChr2, whereas F7GlcT syntenic blocks were observed on OsCh5, ZmChr3, ZmChr8, BdChr1 and EgChr2 (Fig. 5d, e). Closer scrutiny revealed that F5GlcTs found in EgChr1 were arranged in a head-to-tail manner, whereas their paralogues on OsChr7, ZmChr7 and BdChr2 were all separated by putative transposons (Fig. 5f). Further enzyme assays using recombinant proteins also support independent duplication events (Fig. 5f). For instance, both ZmUGT707A8 (from maize) and BdUGT707A15 (from Brachypodium) display similar flavonoids

glycosylation activities with OsUGT707A2 (Fig. 5b). Similarly, in accordance with OsUGT707A4 from rice (Fig. 5b) neither ZmUGT707A18 nor BdUGT707A14 is able to utilize flavonoids as substrates. By contrast, EgUGT707C1, EgUGT707C2 and EgUGT707C3 from palm mainly exhibit flavone 7-O-glucosyltransferase activities (Fig. 5b), showing different specificities from their Gramineae counterparts. When taken together, these data allow us to postulate the hypothetical evolutionary framework for genes in this cluster in the form of the speciation tree based on the NCBI taxonomy database, presented in Fig. 5g.

**Discussion**

Unraveling the natural variation of plant metabolites and the genetic basis underlying this has received increasing interest in recent years[23,46,47]. Analysis of the natural occurring allelic variance not only provides a better understanding of gene function within both ecological and evolutionary contexts[48], but also represents an effective way to determine the function of genes such that they can subsequently be used in gain-of-function breeding approaches. Natural variation of metabolites has been reported to be regulated at transcriptional, translational and posttranslational levels[47,49]. Here we showed that variation of major rice flavonoids is determined by UGT707A2 and UGT706D1, and is regulated at RNA transcript and enzyme activity level, respectively, thus providing an example of multi-layered regulation of natural variation within a single biosynthetic pathway. In addition, the rare allelic variation of the loss-of-function mutation is caused by an InDel that resulted in a frameshift (Fig. 3e, f), compared to less severe variation (which is more common in nature). Consistent with this, close scrutiny revealed that the allelic variation that contributes to the impaired mutant is mostly occurring outside of the enzyme active site region, except one mutant near the flavonoid binding pocket (Supplementary Fig. 9). This mutation (Pro to Leu, no change in amino acid polarity) lies at the turning of a loop into a helix and no difference was observed between the structures harboring either Pro or Leu at this position (Supplementary Fig. 9), suggesting P83L is not crucial for enzyme activity, although this notion remains to be further examined experimentally. Thus, our study is similar to other reported examples[41,42] concerning the cumulative effect of multiple minor mutations outside of the enzyme active site region.

Given the multi-step nature of flavonoid biosynthesis, it is interesting that their variation is mainly determined by tailoring enzymes that catalyze the final step of their biosynthesis. Similar observations were made regarding the genetic architecture of other plant metabolite such as glucosinolates[50–52] and flavonols[23,44] in Arabidopsis, α-tocopherol in tomato[47] and rice secondary metabolism[53–55]. It thus appears likely that decorating enzymes, which are responsible for the final steps of metabolite synthesis generally make a greater contribution to the natural variation of metabolite abundance than early pathway enzymes. One possible explanation of it could be that variation in the activities of these proteins is of lesser consequence to central metabolism—a fact that may benefit the overall fitness of the organism. With this in mind, it is worth noting that tailoring enzymes and/or their products may have quantitative, feedback inhibition, on the upstream biosynthetic steps[27,29,30,56,57], providing a biochemical as well as a genetic explanation for our results.

UV irradiance has a high mutagenic potential given that it is damaging to DNA and RNA[58]. Increasing evidence demonstrates that elevated UV-B radiation is decreasing the yield of many crops[45,59,60], including some cultivars of rice[9,61,62]. To adapt to climatic changes of light quality, plants produce UV-protective phytochemicals. Generally, three types of phytochemicals have

been considered in response to UV-B: ascorbate, hydro-xycinnamates and flavonoids, the role of these phytoprotectants being confirmed by studies using knockout mutants deficient in their biosynthesis[10]. Flavonoids are widely distributed in the plant kingdom and form an important group of plant secondary metabolites it being estimated that flux through the pathway constitutes approximately 20% of the total carbon flux[63]. Our data suggested that OsUGT706D1 and OsUGT707A2 are responsible for the biosynthesis of flavone O-glucosides, which confer enhanced UV-B tolerance in rice. A recent study in Arabidopsis showed that phenylacylated-flavonols, collectively named saiginols, function as UV-B protectants[44]. Collectively and considering the effect of flavonoids on other stresses such as enhanced drought tolerance[64], these two studies thus suggest that increased levels of flavonoids may represent a strategy for stabilizing crop yield under UV-B, drought, or even against the combination of multiple stresses, which frequently occur in the field.

Interestingly, the combination of UGT707A2 and UGT706D1 alleles (RR and AA) is reflected in the geographical distribution of rice cultivars in China, the rest of Asia and even in Europe (Fig. 4g, h and Supplementary Fig. 10d). We found that most cultivars overaccumulating flavone O-glycoside (AA genotype) stem from high irradiance habitats. However, the weak alleles (RR genotype) were also observed in elite cultivars from these regions (Supplementary Data 6). Incorporation of strong alleles (AA genotype) into these cultivars could, therefore, potentially increase their yields. They would additionally be more beneficial in coping with the increased levels of abiotic stress that future crops are anticipated to be exposed to. However, considering the laboratory-UV assay cannot precisely mirror what happens in the field, the true biological gain of the introduction of the AA allele into the RR background still remains to be established.

## Methods

**Plant materials and metabolite profiling.** All plants used in this study were grown in Huazhong Agricultural University, Wuhan. Fern and peat moss were collected on campus and identified according to FRPS (http://frps.eflora.cn/). Potato, citrus, palm, populous and bamboo were obtained from college of Horticulture and Forestry Sciences. Arabidopsis, tobacco and crops were grown in greenhouse. Thirteen plant species leaf samples and tuber of potato were collected using liquid nitrogen with two biological replicate sets. The freeze-dried samples were crushed using a mix mill (MM 400, Ratsch) with a zirconia bead for 1 min at 30 Hz, 100 mg dried power were weighted and extracted overnight at 4 °C with 1.0 mL 70% aqueous methanol containing 0.1 mg $L^{-1}$ lidocaine (internal standard) before analysis using an LC-ESI-MS/MS system[34]. Qualification of metabolites was carried out using a scheduled multiple reaction monitoring method[34]. The relative signal intensities of flavonoids were standardized by firstly dividing them by the intensities of internal standard and then log 2 transforming them to generate the final data matrix. Flavonoids were quantified based on comparison with standards of apigenin, tricin, apigenin 5-O-glucoside and apigenin 7-O-glucoside.

**Nomenclature of UGT-encoding genes.** The nomenclature of full-length UGT-encoding genes that had signature PSPG motifs were obtained (Supplementary Data 7) with the help of the UGT nomenclature committee (http://prime.vetmed. wsu.edu/resources/udp-glucuronsyltransferase-homepage).

**Genome-wide association mapping by gene-based analysis.** Only SNPs with Minor Allele Frequency (MAF) ≥ 0.05 and the number of varieties with the minor allele ≥6 in a (sub) population were used to carry out GWAS. Population structure was modeled as a random effect in LMM using the kinship (K) matrix. We performed GWAS using LMM provided by FaST-LMM program[65]. Instead of calculating the P value for the large number of SNPs, a test of association between the trait and each of the available SNPs within a gene was carried out and the resulting P values and pair-wise correlation coefficients $\gamma$ for all the SNPs are obtained. GATES, a modified of Simes test, was used to combine these available P value to give a gene-base P value as described[38]. The genome-wide significance threshold $(6.6 \times 10^{-8})$ of the gene-based GWAS were determined following Bonferroni correction[66].

**Phylogenetic analysis.** The amino acid sequences of reported UGTs were obtained from NCBI (http://www.ncbi.nlm.nih.gov/). The amino acid sequences of rice UGTs and the related UGTs in this study were extracted from the TIGR

(http://rice.plantbiology.msu.edu/index.shtml), Phytozome 10.3 database (http://phytozome.jgi.doe.gov/pz/portal.html) and Genomsawit Website (http://genomsawit.mpob.gov.my/). The amino acid sequences were aligned using the ClustalW bundled in MEGA 5[67]. The neighbor-joining trees were constructed using MEGA 5 software with all default parameters. The reliability of reconstructed tree was evaluated by bootstrap test with 1000 replicates.

**Recombinant protein analysis and enzyme assay.** The full cDNA of UGTs from Nipponbare (O. sativa L. spp. japonica) were cloned into the pGEX-6p-1 expression vector (Novagen) with a glutathione S-transferase tag. Recombinant proteins were expressed in BL21 (DE3) cells (Novagen) following induction by addition of 0.1 mM isopropy-β-D-thiogalactoside (IPTG) and growing continually for 16 h at 20 °C. Cells were collected and pellets were resuspended in lysis buffer (50 mM Tris-HCl, pH 8.0, 400 mM NaCl). The cells were disrupted by the high pressure cracker and cell debris was removed by centrifugation (14000 g, 1 h). Glutathione Sepharose 4B agarose (GE Healthcare) was added to the supernatant containing the target proteins. After incubation for 1 h, the mixture was transferred into a disposable column and washed extensively with lysis buffer (5 column volumes). Target proteins in collections were confirmed by SDS-PAGE and purified recombinant proteins were selected for enzyme assays and kinetics determination.

The enzyme reactions in vitro assay for glycosyltransferases were performed in a total volume of 100 μl containing 200 μM flavonoid substrates, 1.5 mM UDP-glucose, 5 mM $MgCl_2$ and totally 500 ng purified protein in Tris-HCl buffer (100 mM, pH 7.4) was incubated at 37 °C. After incubating for 20 min, the reaction was stopped by adding 300 μl of ice-cold methanol. The reaction mixture was then filtered through a 0.2 μm filter (Millipore) before being used for LC-MS analysis. HPLC conditions for the analysis of flavonoids were as follows: column, shim-pack VP-ODS (150 L × 4.6); flow rate, 0.8 mL $min^{-1}$; solvent A, 0.04% (by volume) acetic acid in water; solvent B, 0.04% acetic acid in acetonitrile. After injection (40 μL) into a column that had been equilibrated with 5% solvent B (by volume), the column was initially developed isocratically with 5% solvent B, followed by a linear gradient from 5 to 95% solvent B for 20 min. The column was then washed isocratically with 95% solvent B for 2 min, followed by a linear gradient from 95 to 5% solvent B for 0.1 min. The column was isocratically with 5% solvent B for 5 min. The chromatograms were obtained with detection at 330 nm. Peak identification of each component was confirmed using authentic samples and post-run by LC-MS/MS analysis. The amounts of flavonoids were determined from peak integration using authentic samples for calibration.

**Enzyme kinetics.** To determine the kinetic constants of UGTs for flavonoid acceptors, their activities were determined using 0 to 400 μM different flavonoids at a fixed concentration of 1.5 mM UDP-glucose. Flavonoids were purchased from BioBioPha Co., Ltd (www.biobiopha.com/) and UDP-sugars were purchased from Sigma-Aldrich, USA. All Kinetic parameters were calculated using Michaelis-Menten model (Sigma Plot, version 12.5). All the reactions were run in duplicate, and each experiment was repeated twice.

**Rice transformation and qRT-PCR analysis.** An overexpression vector (pJC034) for rice was constructed from the gateway overexpression vector pH2GW7, 35 s promoter of pH2GW7 was replaced by maize ubiquitin promoter. The overexpression constructs of UGT705A2, UGT706C4, UGT706D1, UGT706F1and UGT707A2, were generated by directionally inserting the full cDNAs first into the entry vector pDONR207 and then into the destination vector pJC034 using the Gateway recombination reaction (Invitrogen). The constructs were introduced into Agrobacterium strain EHA105 and then transferred into japonica ZH11. Gene expression for each line was quantified by qRT-PCRusing the relative quantification method. The primers used in this study are presented in Supplementary Data 8. For each construct, at least three independent overexpression plants were selected for the targeted metabolite analyses. The freeze-dried samples were extracted overnight at 4 °C with 1.0 mL 70% aqueous methanol containing 0.1 mgL$^{-1}$ lidocaine (internal standard) before analysis using an LC-ESI-MS/MS system[34].

**UV treatment and NBT staining.** All of the selected flavonoids were prepared in 2 mM concentration and UV irradiation for fluorescence was performed in DNA gel image system (Bio-Rad, 302 nm). All of the seedlings that were used in the UV treatment were grown in the same condition. Four-week old seedlings (OsUGT707A2 overexpression plants, OsUGT706D1 overexpression plants and ZH11, 24 plants each, and three repeats) were treated with UV-B radiation for 10 h, and the survival rates were calculated. The plants with normal green and unwithered leaves were considered as survived plants. For NBT staining, leaves of survived plants were harvested after 5 h UV-B radiation and vacuum infiltrated with 0.1% nitroblue, 10 mM sodium azide in 10 mM phosphate buffer, pH 7.6. Samples were incubated for 1 h at room temperature in the light. To remove chlorophylls, the stained samples were transferred to 95% ethanol and incubated at 80 °C for 15 min.

**Agriculture phenotype.** OsUGT707A2 overexpression plants and WT (ZH11 variety) were grown in Wuhan, Hubei province during May to August at 2015. These plants were also grown in Haikou, Hainan province during December to

March from 2015 and 2016. Harvested paddy rice was dried and stored at room temperature for at least 3 months before measurement. 100 fully filled grains were randomly selected and weight within each line with three repeats. The values were averaged and used as the measurements for hundred-grain weight.

**Data availability**. The authors declare that all data supporting the findings of this study are available within the manuscript and its supplementary information files or are available from the corresponding author upon request.

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

## Acknowledgements

This work was supported by the National Science Fund for Distinguished Young Scholars (No. 31625021), the Ministry of Science and Technology of China (2016YFD0100500), the State Key Program of National Natural Science Foundation of China (No. 31530052).

## Author contributions

J.L. conceived the project and supervised this study. M.P., R.S., A.G. and H.S. performed most of the experiments; S.S., L.L., D.L., E.N. and X.L. participated in the material preparation; M.P., S.S., J.Z., S.W. and J.L. carried out the metabolite analyses; Z. Z performed the GWAS analysis. M.P., T.T., A.F. and J.L. analyzed the data; M.P., T.T., A. F. and J.L. wrote the paper. All of the authors discussed the results and commented on the manuscript.

## Additional information

**Competing interests:** The authors declare no competing financial interests.

