## [Peer Review File · Nature Communications]

Reviewers' comments:

Reviewer #1 (Remarks to the Author):

The authors use a previous dataset to identify and validate a couple of flavone UGT's to add to the rapidly growing massive list of secondary metabolite genes in plants. They show that there is an effect of over-expressing this gene on short-term UV-B exposure and a correlation of the SNPs to field UV-B levels. This is an interesting beginning of what could be an interesting project.

In the manuscript, the authors over-claim that have proven that these alleles contribute to natural variation in resistance to UV-B. They have some indication that there is a correlation and that the transgenic line provides tolerance. But the tolerance assay is not the same as the field conditions. In the field, the plants germinate in the presence of the UV-B and it stays at the same level throughout with diurnal and cloud driven oscillations. There is never a day where the field grown plant does not experience the UV-B. As such, the tolerance assay is not directly linked to the field assay. The authors should properly caveat in each and every instance that they simply have an association.

I am confused by this sentence about looking for SNPs in the two UGTs “We identified a total of 20 and 39 SNPs, respectively, which were above the cut-off of a 5% false discovery rate (Fig. 3a and 3b).”. This isn't a statistical analysis but is instead a simple SNP discovery, correct? If so, then why do you expect 5% SNPs by random chance, 5% FDR,? Or was this a GWAS analysis? This is not very clear.

Also, the authors should caveat the SNP database as this database could have significant issues with larger indels, inversions or other larger changes. As such, they may not have all the possible polymorphisms. As such, they cannot really claim that the two polymorphisms on line 245 are the only possibilities. Additionally, the silent SNPs could affect RNA levels or splicing even if not in the splice acceptors.

On lines 263-266, it is interesting that there is a yellow fluorescence emitted but this does not prove that it is more efficient in reducing UV-B damage. Especially as it is not clear in Figure 4A if this is just UV-B or also UV-A/C.

On lines 370-372, the authors go from saying flavones are ubiquitous to saying they are missing

in Brassica. That would mean that they are not ubiquitous. This should be corrected.

In Figure 1, is the phylogenetic tree including all the UGTs or just a subset? If so, how was this subset chosen? If the authors really want to discuss duplication and neofunctionalization, then they need to include all possible genes in the region of the tree involving this process. Right now, it is not clear what is occurring as the branches of the neofunctionalized gene members are often as long as between monocot and dicot genes in the rest of the tree. This is further developed in Figure 5 but this figure is so tiny as to be illegible for the key points. It seems to be saying that this “neofunctionalization” event occurred early in the monocots (Figure 5F) but then 5G talks about duplication and deletion without commenting about activity. The key to this claim is to link activity with genomic arrangement. From Figure 5F, all I really see is that there were four old activities (by color) and then they have been maintained in most monocot lineages but there is no real source for this neofunctionalization. This is key because most of the papers that find neofunctionalization for driving diversity are looking at within species duplication and neofunctionalization, not pre-family events. My sole request on this is to discuss what is actually in the data rather than conflating with the broader topic of diversity evolution. Right now the paper makes it sound like this is duplication and diversification in rice leading to diversity.

For the GWAS, the authors provide almost no information on the level of replication or experimental design. This needs to be rectified. If this is in another paper where these gene to trait associations have previously been discussed, then this also needs to be clarified.

As a side note, it seems that these polymorphisms have already been efficiently incorporated into rice breeding across China and India. So it is not clear how knowing these genes will improve breeding for stress resistance.

Reviewer #2 (Remarks to the Author):

Peng et al. explore the importance and evolution of flavone glucosylation in rice. In contrast to the intensely studied flavonol and anthocyanin glycosylation (mostly, but not exclusively studied in Arabidopsis), the conjugation of flavones has not been addressed in a comparable manner. Yet, flavone (glycosides) are important constituents of monocot crops. The authors present a well-designed and straightforward study linking UGT isoforms to the abundance of flavone-hexosides, analyze their activities and link this information with UV-B tolerance and geographical distribution as well as to the evolution of flavone glycosylation.

An mGWAS of >500 rice accessions revealed four highly significant chromosomal regions linked to flavone 5-O- and flavone-7-O-hexosides. Interestingly, all regions harbored UGT isoforms, which were further analyzed in vitro identifying flavone glucosyltransferases. Allelic

versions with differential activity towards flavones supported the evolution of these isoform and linked it to UV-B sensitivity (in lab and field experiments). Thereby it was also clearly shown that flavones glucosides are important for

Both the strategy of the study (also interesting to a broader community) and the findings concerning flavone glucosylation will be of high interest to the plant research community.

Nevertheless, there are a few issues that should be addressed by the authors:

1) Discussion is a little bit disappointing after the nice and clear result section, since it largely reviews the results (in particular lines 381-392) and does not indicate new aspects or conclusions which were even laid out already in the result section. E.g. one could consider (i) the importance of the allele in breeding compared to the currently found geographical distribution. Suppl. Fig. 7 seems to indicate that the "right" cultivars are used in regions exposed to higher UV irradiance? I.e. finally, how much benefit could be achieved in addition: a question that is justified for a manuscript where the abstract ends by pointing to "an important step towards stabilizing the yield of the world's most important grass crop."

(ii) I find it highly interesting that the conjugating UGT enzymes make the difference and not primarily changes in the backbone biosynthesis. Is there any precedence? Is there any evolutionary "stratagem" or "advantage" of such a modulation? In fact, it also reminds me to a study where the glucosylation of flavonols (or its loss-of-function) in Arabidopsis had an impact on the biosynthesis of the flavonol backbone.

(iii) And more specifically regarding the R vs. A alleles: where are the changes located? Since the UGT structure is quite well known and these new isoforms can be modeled, one could (possibly) deduce some ideas or comments on the mutations and their impact.

2) Line 201: are there galactosides in rice?

3) Line 307: why do you address the aglycones, these have not been addressed (and may not even exist as such in considerable quantities, since they will not be soluble etc.).

Minor issues:

1) Line 162: fused to the "N"-terminus (not "C"), if the sentence is formulated like it is.

2) Line 194: "5-O" should be canceled, since m/z does not allow this specification (although the text is formally correct: "consistent with the mass calculation of tricin-5-O-glucose")

3) Lines 287-290: The UV-protective effect can be deduced from their absorption spectra? Why is this not mentioned?

4) Typos etc.

Line 78: "nor": please check syntax of the sentence?

Line 193: "ratio"

Line 209: "flavone" only

Line 260: "it" remains unclear whether flavones confer UV resistance in plants

Line 263: "flavonols"

Line 323: "proven"

Line 358: cancel "of"

Line 361: "deficiency"

Suppl. Fig. 7c: "irradiance" (x-axis)

Reviewer #1 (Remarks to the Author):

The authors use a previous dataset to identify and validate a couple of flavone UGT's to add to the rapidly growing massive list of secondary metabolite genes in plants. They show that there is an effect of over-expressing this gene on short-term UV-B exposure and a correlation of the SNPs to field UV-B levels. This is an interesting beginning of what could be an interesting project.

In the manuscript, the authors over-claim that have proven that these alleles contribute to natural variation in resistance to UV-B. They have some indication that there is a correlation and that the transgenic line provides tolerance. But the tolerance assay is not the same as the field conditions. In the field, the plants germinate in the presence of the UV-B and it stays at the same level throughout with diurnal and cloud driven oscillations. There is never a day where the field grown plant does not experience the UV-B. As such, the tolerance assay is not directly linked to the field assay. The authors should properly caveat in each and every instance that they simply have an association.

Response: As you mentioned, we agree with that the UV tolerance assay is not the same as the field trial and not directly linked to the field condition. We have made corresponding changes as you suggested in order to correctly caveat this (**lines 314-316, 420-422**). Thank you for bringing to our attention this important detail.

I am confused by this sentence about looking for SNPs in the two UGTs “We identified a total of 20 and 39 SNPs, respectively, which were above the cut-off of a 5% false discovery rate (Fig. 3a and 3b)”. This isn't a statistical analysis but is instead a simple SNP discovery, correct? If so, then why do you expect 5% SNPs by random chance, 5% FDR? Or was this a GWAS analysis? This is not very clear.

Response: 5% is not statistical analysis. It is included in GWAS analysis that indicates the value of minor allele frequency. In our analysis, only SNPs with minor allele frequency ≥ 0.05 were used to carry out GWAS (also described in methods, lines 447-449) and could be used in our further analysis. We have modified this sentence (**lines 227-228**) for clarity and hope this has dispelled the confusion.

Also, the authors should caveat the SNP database as this database could have significant issues with larger InDels, inversions or other larger changes. As such, they may not have all the possible polymorphisms. As such, they cannot really claim that the two polymorphisms on line 245 are the only possibilities. Additionally, the silent SNPs could affect RNA levels or splicing even if not in the splice acceptors.

Response: We agree with that the two polymorphisms we found are not the only possibilities. They are among the possible ones. We also agree with your statements that other large changes or silent SNPs are possible. Since we have not sequenced the whole locus, the effects of unidentified polymorphisms cannot be ruled out, nor the possible roles of the “silent” ones as they may effect RNA levels or splicing. We have modified our description accordingly (**lines 247-250**) and again thank you for the insightful comment which allowed us to state this more carefully.

On lines 263-266, it is interesting that there is a yellow fluorescence emitted but this does not prove that it is more efficient in reducing UV-B damage. Especially as it is not clear in Figure 4A if this is just UV-B or also UV-A/C.

Response: We agree with that a yellow fluorescence cannot prove it is more efficient in reducing UV-B damage. Therefore, we have removed this sentence. We did the assay only under UV-B irradiation, and the results from Figure 4A showed yellow fluorescence emit under this condition (302nm). We have made it clear in current manuscript (**lines 271-274**) as well as in the legend to Figure 4a.

On lines 370-372, the authors go from saying flavones are ubiquitous to saying they are missing in Brassica. That would mean that they are not ubiquitous. This should be corrected.

Response: Sorry for this discrepancy. We have changed the term “ubiquitous” to “widely distributed”.

In Figure 1, is the phylogenetic tree including all the UGTs or just a subset? If so, how was this subset chosen? If the authors really want to discuss duplication and neofunctionalization, then they need to include all possible genes in the region of the tree involving this process. Right now, it is not clear what is occurring as the branches of the neofunctionalized gene members are often as long as between monocot and dicot genes in the rest of the tree. This is further developed in Figure 5 but this figure is so tiny as to be illegible for the key points. It seems to be saying that this “neofunctionalization” event occurred early in the monocots (Figure 5F) but then 5G talks about duplication and deletion without commenting about activity. The key to this claim is to link activity with genomic arrangement. From Figure 5F, all I really see is that there were four old activities (by color) and then they have been maintained in most monocot lineages but there is no real source for this neofunctionalization. This is key because most of the papers that find neofunctionalization for driving diversity are looking at within species duplication and neofunctionalization, not

pre-family events. My sole request on this is to discuss what is actually in the data rather than conflating with the broader topic of diversity evolution. Right now the paper makes it sound like this is duplication and diversification in rice leading to diversity.

Response: Our phylogenetic tree presented in Fig.1D includes all the UGTs with typical activities, covering all reported flavonoid glycosylation functions (CGT, 3'GT, 3GT, 5GT, 7GT and GGT). We appreciate your opinions on gene duplication and neofunctionalization. Further, we came back to our data and realized our present results suggested a distinct evolution of F5GT and F7GT, but did not provide direct evidence for gene duplication and neofunctionalization within species. We agree with your suggestion to avoid discussing with the broader topic of diversity evolution. The second reviewer also suggested us to rewrite our discussion section. To this end, discussion on duplication and neofunctionalization has been eliminated and new aspects were included in current manuscript (**lines 363-422**).

For the GWAS, the authors provide almost no information on the level of replication or experimental design. This needs to be rectified. If this is in another paper where these gene to trait associations have previously been discussed, then this also needs to be clarified.

Response: We have added the replication information and experimental design in **lines 127-129**. The data of metabolite traits and genotypes are obtained in our previous studies in rice (Chen et al., Nature Genetics, 2014 and Chen et al., Nature Communications, 2016). In both studies, metabolite profiling data were obtained from two independent biological sample sets. In this study, we performed gene-based GWAS analysis and identified novel loci (i.e. L1, L2, L3 as shown Fig. 1b) that have been detected repeatedly in each of the studies. We subsequently characterized all the loci by further molecular and biochemical studies, resulting in the identification a total of 12 flavonoid glucosyltransferases and the elucidation of their biochemical and biological roles.

As a side note, it seems that these polymorphisms have already been efficiently incorporated into rice breeding across China and India. So it is not clear how knowing these genes will improve breeding for stress resistance.

Response: Our results (Figure 4g and Supplementary Fig. 7d) showed that many but not all major lines from the high irradiance region harbor strong allele combination (AA), therefore it is possible to introduce strong allele combination (AA) into the cultivars with weak/moderate allele combinations (RR or RA), for example some elite

lines in Yunnan-Kweichow Plateau, the East of China, Philippines and Vietnam. Our extra data (provided in **Supplementary Table 8**) summarized all cultivar information, additionally with data of two-year thousand grain weight (TGW, an important factor in yield). It clearly shows which elite lines we can select to improve breeding for stress resistance.

Reviewer #2 (Remarks to the Author):

Peng et al. explore the importance and evolution of flavone glucosylation in rice. In contrast to the intensely studied flavonol and anthocyanin glycosylation (mostly, but not exclusively studied in Arabidopsis), the conjugation of flavones has not been addressed in a comparable manner. Yet, flavones (glycosides) are important constituents of monocot crops. The authors present a well-designed and straightforward study linking UGT isoforms to the abundance of flavone-hexosides, analyze their activities and link this information with UV-B tolerance and geographical distribution as well as to the evolution of flavone glycosylation.

An mGWAS of >500 rice accessions revealed four highly significant chromosomal regions linked to flavone 5-O- and flavone-7-O-hexosides. Interestingly, all regions harbored UGT isoforms, which were further analyzed in vitro identifying flavone glucosyltransferases. Allelic versions with differential activity towards flavones supported the evolution of these isoform and linked it to UV-B sensitivity (in lab and field experiments). Thereby it was also clearly shown that flavones glucosides are important for both the strategy of the study (also interesting to a broader community) and the findings concerning flavone glucosylation will be of high interest to the plant research community.

Nevertheless, there are a few issues that should be addressed by the authors:

1) Discussion is a little bit disappointing after the nice and clear result section, since it largely reviews the results (in particular lines 381-392) and does not indicate new aspects or conclusions which were even laid out already in the result section. E.g. one could consider (i) the importance of the allele in breeding compared to the currently found geographical distribution. Suppl. Fig. 7 seems to indicate that the "right" cultivars are used in regions exposed to higher UV irradiance? I.e. finally, how much benefit could be achieved in addition: a question that is justified for a manuscript where the abstract ends by pointing to "an important step towards stabilizing the yield of the world's most important grass crop."

(ii) I find it highly interesting that the conjugating UGT enzymes make the difference and not primarily changes in the backbone biosynthesis. Is there any precedence? Is

there any evolutionary "strategy" or "advantage" of such a modulation? In fact, it also reminds me to a study where the glucosylation of flavonols (or its loss-of-function) in *Arabidopsis* had an impact on the biosynthesis of the flavonol backbone.

(iii) And more specifically regarding the R vs. A alleles: where are the changes located? Since the UGT structure is quite well known and these new isoforms can be modeled, one could (possibly) deduce some ideas or comments on the mutations and their impact.

Response: we have performed protein modeling and the result is shown in **lines 260-262**. According to your suggestion, we have reshaped our discussion section that now focused on (i) natural variation of UGTs and where are the polymorphisms' locations and their impacts, and, (ii) the evolution advantage that decorative modulation in which UGT enzymes make the difference and not primarily changes in the flavone backbone is additionally discussed, (iii) the importance of the allele in breeding programs was expanded.

2) Line 201: are there galactosides in rice?

Response: there are no flavonoid galactosides in rice. We use UDP-galactose as sugar donor only to test the sugar donor specificity. We have modified this sentence to "in terms of sugar donor, catalytic activities were only observed with UDP-glucose but not with UDP-galactose" (**lines 202-204**). These observations were consistent with the absence of flavonoid galactosides in rice and the previous studies (Yonekura-Sakakibara et al., *Plant Cell*, 2008; Noguchi et al., *Plant Cell*, 2009; One et al., *Plant Cell*, 2010).

3) Line 307: why do you address the aglycones, these have not been addressed (and may not even exist as such in considerable quantities, since they will not be soluble etc.).

Response: we have modified this sentence according to your suggestion and the sentence is now "Taken together, the abundance of flavone *O*-glucosides appears related to adaptation to UV-B irradiance" (**lines 313-314**).

Minor issues:

1) Line 162: fused to the "N"-terminus (not "C"), if the sentence is formulated like it is.

Response: we have modified this sentence according to your suggestion.

2) Line 194: "5-*O*" should be canceled, since m/z does not allow this specification

(although the text is formally correct: "consistent with the mass calculation of tricetin 5-*O*-glucose")

Response: we have removed "5-*O*" according to your suggestion

3) Lines 287-290: The UV-protective effect can be deduced from their absorption spectra? Why is this not mentioned?

Response: We agree with that the UV-protective effect can be deduced from absorption spectra (at a wavelength of 280-350nm) and we have mentioned it in our manuscript (**lines 266-268**).

4) Typos etc.

Line 78: "nor": please check syntax of the sentence?

Response: We have corrected this sentence as "For example, few flavone-active UGTs have been characterized and the natural genetic variation of rice flavones are largely unknown, as is *in planta* biological function of glycosylated flavones." (**lines 76-79**)

Line 193: "ratio"

Line 209: "flavone" only

Line 260: "it" remains unclear whether flavones confer UV resistance in plants

Line 263: "flavonols"

Line 323: "proven"

Line 358: cancel "of"

Line 361: "deficiency"

Suppl. Fig. 7c: "irradiance" (x-axis)

Response: Thanks for your kind suggestions. We have made the corresponding changes.

Reviewers' comments:

Reviewer #1 (Remarks to the Author):

The authors have improved the manuscript but there are still key issues that were either incorrectly adjusted or added anew that need correction to allow this reviewer to understand what has or has not been done. Critically, it is not clear how the GWAS was done or integrated.

The correction on lines 227-228 is actually not clarifying. It is useful to get rid of the term false discovery rate when the authors meant minor allele frequency. But now the new line has no mention of minor allele frequency. Instead, the new line seems to imply that the 50 SNPs are the total number of SNPs without even mentioning that there is a minor allele frequency cutoff. Additionally it is not clear how many of these SNPs are or are not significantly associated with the trait in question. Further, an indel and a SNP are not equivalent and the two seem to be conflated on lines 227-228. The authors must specify exactly what has been done and be very clear that they used MAF cutoffs. Further, they must state what are SNPs and what are indels in this line. Finally, they need to state which of the SNPs if any are significant. Figure 3d does not provide a test of significance while using population structure in the same model.

I am unclear on the response about including all of the UGTs with the typical activities? My concern was that all UGTs should be included based on sequence homology and not merely based on proven enzymatic activity. Large gene families typically have a great number of genes with no proven activity and these genes are as important to phylogenetic interpretation as are the ones with proven activity. Phylogenies should be explicit about what is in the tree and what is not in the tree. Currently, I am still unclear about the phylogeny.

I also have some further concerns about the statistics. In the following “The relative signal intensities of flavonoids were normalized by firstly dividing them by the intensities of internal standard and then log₂ transforming them for further normalization to improve the normality”. The first normalized is not a normalization, it is merely a standardization to the standard. The log₂ is just a scale adjustment in this case as there is no reason to assume that the accumulation of these compounds across species would fit a normal Gaussian curve. Further, I did not see any statistics for significance applied to the cross species data.

After re-reading the Nature Communications and Nature Genetics papers and the manuscripts materials and methods paper, I am still unsure how the GWA was conducted. The Nature Communications paper talks about data from two years while the Nature Genetics paper talks about data from two fields. This then somehow yields a single Manhattan plot in Figure 1b. Does Figure 1b show four overlapping, two overlapping or 1 GWA result? If either four or two, then

there has to be some way to differentiate the result from the different results. If the four results are combined statistically somehow, it is not clear how they are combined. Integrated does not specify how the integration occurred.

The new discussion has some significant issues. For example, the senior author solely cites himself for key claims that were established in other systems first. This is illustrated by “Similar observations were made for other plant metabolites such as α -tocopherol⁵², dihydroxybenzoic acid glycosides⁵³ and phenolamides⁵⁴, and even for human metabolites⁵”. It would seem that the glucosinolate natural variation by Mithen, Kroymann, Kliebenstein and others actually illustrated this nearly a decade prior to these citations. Further, the work by Saito and colleagues in *Arabidopsis*, equally demonstrated this concept of most natural variation in flavonols was in the tailoring enzymes. Equally, work by Fernie and colleagues in *Arabidopsis* has shown this about flavonols as well. This lack of citation to previous evidence is equally true of the feedback claim in the next sentence of the discussion as this was also shown in the work by the above authors about both glucosinolates and flavonols. The authors should provide appropriate citation for all work within the field and not simply their own.

Reviewer #2 (Remarks to the Author):

The authors dealt with all issues raised by both reviewers. However, in particular the responses to reviewer 1 seemed to weaken the paper at first glance when I read the authors' response. This impression stems from the fact that they basically agreed with several critical questions and acted only by adding remarks of caution which will weaken some of their important conclusions.

When re-reading the critical sections, however, I found that the evidence is still nicely presented. In some cases, the cautionary remarks may be even overstating: e.g. the correlation of flavone glucosides with functional OSUGT706D1 and OsUGT707A2 is shown by several lines of evidence: SNPs, InDel, mRNA expression level, protein activities. Therefore, the note added in 248-250 is of course correct, but it might not even be necessary to seriously invoke additional potential reasons with the massive evidence already presented.

Another aspect is field vs. laboratory UV tolerance: this is indeed very different, but this has now been correctly addressed. Nevertheless, with the many cultivars assessed, there is at least a good correlation of the flavone glucoside-favoring alleles and global UV impact. Thus, this supports this work as an interesting publication.

Nevertheless, there are a few aspects that still should be addressed:

1) Discussion: in some paragraphs I had the impression as they were written rather quickly

(including several typos or mistakes in English grammar, e.g. line 378-379, 387, 412, 415, 416). In terms of contents, the issues are only roughly addressed, which after all may be fine, since the results were clear. However, notions like "suggests a possible evolutionary advantage of the knockdown lines in comparison to knockout lines" (lines 376-378) sound somewhat strange and are at most page filling.

2) Discussion: the final remark on the introgression of AA alleles into cultivars used in Southern regions is fine, but in total a rather weak and short discussion (only) of the strong last sentence of the abstract claiming or promising that these alleles "will likely represent an important step towards stabilizing the yield of the world's most important grass crop." At least I would expect more from such a statement.

3) Results. line 307: if I get it right, then the allele combinations have been mixed here: AA (not RR) for the high flavone glucoside varieties, in agreement with the figure.

4) Results (lines 260-264) and Discussion (lines 378-382): For sure, mutations outside the active center may impact on the enzyme's activity. However, there is a massive mutation, Leu83Pro, in close vicinity to the active center. Since their impact has not been tested separately, a notion that the "allelic variation that contributes to the knockdown, is almost exclusively distributed outside the enzyme activity center" is not substantiated by experimental evidence.

4) Discussion, minor aspect: line 392: since only case is cited, I would suggest to phrase it as "may have quantitative feed-back effect ..." instead of "usually".

Reviewers' comments:

Reviewer #1 (Remarks to the Author):

1. The authors have improved the manuscript but there are still key issues that were either incorrectly adjusted or added a new that need correction to allow this reviewer to understand what has or has not been done. Critically, it is not clear how the GWAS was done or integrated.

Response: In this study, our GWAS data in rice leaf were obtained from two independent replicates in one year and independent GWAS analysis was performed for each replicate. Our GWAS data in rice seeds were obtained from four independent replicates (two replicates × two years) and independent GWAS analysis was performed for each replicate. We found that some significant associations can be detected repeatedly in leaves (both duplicates, **Supplementary Fig. 3a**) and/or in seeds (all four replicates, **Supplementary Fig. 3b**). For example, Locus 1 was repeatedly detected in both repeat 1 and repeat 2 from the leaf for apigenin 7-*O*-glucoside. Similarly, this locus was also detected in all four repeats from the seed for chrysoeriol 7-*O*-glucoside. Therefore, these loci were chosen for further analysis. We overlap the Manhattan plots in order to show 4 main loci (L1-L4) in Figure 1 and have explained this in the Figure legend. We hope this has dispelled the confusion.

2. The correction on lines 227-228 is actually not clarifying. It is useful to get rid of the term false discovery rate when the authors meant minor allele frequency. But now the new line has no mention of minor allele frequency. Instead, the new line seems to imply that the 50 SNPs are the total number of SNPs without even mentioning that there is a minor allele frequency cutoff. Additionally it is not clear how many of these SNPs are or are not significantly associated with the trait in question. Further, an indel and a SNP are not equivalent and the two seem to be conflated on lines 227-228. The authors must specify exactly what has been done and be very clear that they used MAF cutoffs. Further, they must state what are SNPs and what are indels in this line. Finally, they need to state which of the SNPs if any are significant. Figure 3d does not provide a test of significance while using population structure in the same model.

Response: We have modified our statements and clarified that we use MAF cutoff ($MAF \geq 5\%$) in searching all SNPs and InDels (**lines 230-233 and lines 256-259**). We have provided supplemental tables that contain allele frequency and *P*-value for all the SNPs found in the promoter and UTR regions (**Supplementary Tables 6 and 8**). Accordingly, we modified our figure 3 to show the most significant SNPs within UGT707A2 and UGT706D1 (**Figs. 3a and 3f**). For the InDel, we also clarified that it is of low frequency and confirmed by re-sequencing ($MAF=0.629\%$, **lines 246-247**). We have also provided the test of significance for Fig. 3d (now as **Fig.3c**) as you requested.

3. I am unclear on the response about including all of the UGTs with the typical activities? My concern was that all UGTs should be included based on sequence homology and not merely based on proven enzymatic activity. Large gene families typically have a great number of genes with no proven activity and these genes are as important to phylogenetic interpretation as are the ones with proven activity. Phylogenies should be explicit about what is in the tree and what is not in the tree. Currently, I am still unclear about the phylogeny.

Response: Thanks for your suggestion. Here we present a modified phylogenetic tree that contains our candidate genes and their homologs. These homologs were selected as described below: we choose representative UGT gene encoding enzymes from each locus as bait (for instance, UGT706D1 in L1, UGT705A1 in L2, UGT706C4 in L3, and UGT707A2 in L4) and perform protein BLAST in the NCBI database with default settings across species. By using the cutoff (40% identities), we found no hits from the ancestral plants (such as *Selaginella moellendorffii* and *Physcomitrella patens*) and hits from representative monocot and dicot species (such as *Zea mays*, *Triticum aestivum*, *Populus trichocarpa* and *Vitis vinifera*) were used for phylogenetic analysis together with the reported UGTs (3GTs, 5GTs, 7GTs, CGTs and GGTs). Using the sequences mentioned above, a neighbor-joining tree was constructed by using MEGA5 (**Supplementary Fig. 4**). This tree showed that UGTs from locus 2 (in orange color) are close to the 7GT and 3'GT clades, whereas UGTs from loci 1, 3 and

4, and their homologs are distinct from the reported UGTs and form a new clade. These observations are consistent with our phylogenetic tree presented in Fig.1d. Meanwhile, there are two subclades within this new clade. One subclade (include UGTs from L1 and L3) contains homologs from monocots and eudicots. These observations are consistent with our phylogenetic tree in Figs. 5a and 5b. Given that too many genes are crowded here in this large family and this phylogenetic tree is too big to properly fit in our Fig.1, we place this tree in the supplementary material (**Supplementary Fig. 4**) as an extra support.

4. I also have some further concerns about the statistics. In the following “The relative signal intensities of flavonoids were normalized by firstly dividing them by the intensities of internal standard and then log 2 transforming them for further normalization to improve the normality”. The first normalized is not a normalization, it is merely a standardization to the standard. The log2 is just a scale adjustment in this case as there is no reason to assume that the accumulation of these compounds across species would fit a normal Gaussian curve. Further, I did not see any statistics for significance applied to the cross species data.

Response: We agree with that our data processing is standardization rather than normalization. We have made the corresponding change (**lines 449-450**). Further, we have provided the test of significance across species (eg. rice, potato, hullless barley, populus, wheat and tobacco) as examples (**Supplementary Table 2**). The statistical results showed that contents of flavone glycosides are significantly more highly abundant in rice than in potato, acylated flavonoid glycosides are significantly different between hullless barley and populus, and tricin-lignans accumulated at significantly higher levels in monocots, which is consistent with our description (**lines 103-109 and lines 114-119**) and Fig. 1a.

5. After re-reading the Nature Communications and Nature Genetics papers and the manuscripts materials and methods paper, I am still unsure how the GWA was conducted. The Nature Communications paper talks about data from two years while

the Nature Genetics paper talks about data from two fields. This then somehow yields a single Manhattan plot in Figure 1b. Does Figure 1b show four overlapping, two overlapping or 1 GWA result? If either four or two, then there has to be some way to differentiate the result from the different results. If the four results are combined statistically somehow, it is not clear how they are combined. Integrated does not specify how the integration occurred.

Response: Figure 1b shows the overlapping Manhattan plot from the leaf and the seeds. As mentioned in our response to question 1, in this study, our GWAS data in rice leaf were obtained from two independent replicates in one year and independent GWAS analysis was performed for each replicate. Our GWAS data in rice seeds were obtained from four independent replicates (two replicates \times two years) and independent GWAS analysis was performed for each replicate. Loci that can be detected repeatedly in leaves and/or in seeds were chosen for further analysis. We overlap the Manhattan plots to show 4 main loci (L1-L4) in one figure and have now clearly stated this in the Figure legend. We also provide the original Manhattan plots from the 6 individual replicates in **Supplementary Fig. 3** – there the reviewer/reader can see that these loci are conserved across the experiments. Also please note the extremely high LOD scores also provide additional confidence in these data.

6. The new discussion has some significant issues. For example, the senior author solely cites himself for key claims that were established in other systems first. This is illustrated by “Similar observations were made for other plant metabolites such as α -tocopherol⁵², dihydroxybenzoic acid glycosides⁵³ and phenolamides⁵⁴, and even for human metabolites⁵⁵”. It would seem that the glucosinolate natural variation by Mithen, Kroymann, Kliebenstein and others actually illustrated this nearly a decade prior to these citations. Further, the work by Saito and colleagues in *Arabidopsis*, equally demonstrated this concept of most natural variation in flavonols was in the tailoring enzymes. Equally, work by Fernie and colleagues in *Arabidopsis* has shown this about flavonols as well. This lack of citation to previous evidence is equally true of the feedback claim in the next sentence of the discussion as this was also shown in

the work by the above authors about both glucosinolates and flavonols. The authors should provide appropriate citation for all work within the field and not simply their own.

Response: Thanks for your kind suggestions. Many of these works were cited in the previous version of our manuscript however we are happy to re-add them. Natural variation of glucosinolates (work by Kliebenstein and colleagues) and flavonols in Arabidopsis (work by Fernie colleagues) are good examples here and we have added the citations in our current manuscript (**lines 394-395**). The work by Saito and colleagues in rice secondary metabolism are also included (**line 396**).

Reviewer #2 (Remarks to the Author):

The authors dealt with all issues raised by both reviewers. However, in particular the responses to reviewer 1 seemed to weaken the paper at first glance when I read the authors' response. This impression stems from the fact that they basically agreed with several critical questions and acted only by adding remarks of caution which will weaken some of their important conclusions.

When re-reading the critical sections, however, I found that the evidence is still nicely presented. In some cases, the cautionary remarks may be even overstating: e.g. the correlation of flavone glucosides with functional OSUGT706D1 and OsUGT707A2 is shown by several lines of evidence: SNPs, InDel, mRNA expression level, protein activities. Therefore, the note added in 248-250 is of course correct, but it might not even be necessary to seriously invoke additional potential reasons with the massive evidence already presented.

Another aspect is field vs. laboratory UV tolerance: this is indeed very different, but this has now been correctly addressed. Nevertheless, with the many cultivars assessed, there is at least a good correlation of the flavone glucoside-favoring alleles and global UV impact. Thus, this supports this work as an interesting publication.

Response: We thank the reviewer for their highly positive comments we have chosen to leave in the caveat about unidentified polymorphisms since as they state it is factually correct and we felt that such explicit statements were wanted following the

first round of reviews. However, we do not feel strongly either way and would thus be happy to delete this sentence if asked to do so. Similarly, for the comment regarding the discrepancy between greenhouse and field since this is also factually true we merely toned this down. Again we would be happy to delete this sentence if asked to do so.

Nevertheless, there are a few aspects that still should be addressed:

1) Discussion: in some paragraphs I had the impression as they were written rather quickly (including several typos or mistakes in English grammar, e.g. line 378-379, 387, 412, 415, 416). In terms of contents, the issues are only roughly addressed, which after all may be fine, since the results were clear. However, notions like "suggests a possible evolutionary advantage of the knockdown lines in comparison to knockout lines" (lines 376-378) sound somewhat strange and are at most page filling.

Response: We have checked our manuscript and corrected the mistakes in English grammar as you mentioned (**lines 385-387, 423-425 and 427-428**). We have deleted the sentence "suggests a possible evolutionary advantage of the knockdown lines in comparison to knockout lines" as you suggested and made the corresponding change (**lines 382-384**).

2) Discussion: the final remark on the introgression of AA alleles into cultivars used in Southern regions is fine, but in total a rather weak and short discussion (only) of the strong last sentence of the abstract claiming or promising that these alleles "will likely represent an important step towards stabilizing the yield of the world's most important grass crop." At least I would expect more from such a statement.

Response: As you mentioned, the last sentence in our abstract is too strong. Therefore, we have changed it to "adoption of the positive alleles of these genes into breeding programs will likely represent a potential strategy aim at producing stress-tolerant plants" (**line 27**).

3) Results. line 307: if I get it right, then the allele combinations have been mixed

here: AA (not RR) for the high flavone glucoside varieties, in agreement with the figure.

Response: Thanks for your kind suggestions. We are sorry to mix the combination here and we have corrected the mistake in our current manuscript (**lines 315-316**).

4) Results (lines 260-264) and Discussion (lines 378-382): For sure, mutations outside the active center may impact on the enzyme's activity. However, there is a massive mutation, Leu83Pro, in close vicinity to the active center. Since their impact has not been tested separately, a notion that the "allelic variation that contributes to the knockdown, is almost exclusively distributed outside the enzyme activity center" is not substantiated by experimental evidence.

Response: As you suggested, Leu83Pro is located in the flavonoid binding pocket and we should not ignore its effect. Given that both Leucine and Proline are nonpolar amino acid, we have modified our description in **lines 385-389** "Consistent with this, close scrutiny revealed that the allelic variation that contributes to the impaired mutant is mostly occurring outside of the enzyme active site region, except one mutant near the flavonoid binding pocket (Supplementary Fig. 9). The effect of this mutant (Pro to Leu cause no change in amino acid polarity) on enzymatic catalysis remains to be further examined".

4) Discussion, minor aspect: line 392: since only case is cited, I would suggest to phrase it as "may have quantitative feed-back effect ..." instead of "usually".

Response: Thanks for your kind suggestions. We have made the corresponding change (**line 402**).

Reviewers' Comments:

Reviewer #1 (Remarks to the Author):

The authors have largely addressed my concerns. I am now able to understand how the GWA was conducted and what the results are indicating.

Reviewer #2 (Remarks to the Author):

From my point of view, the responses are satisfying.

There is only one minor remark to be considered in the discussion (line 389 with respect to Suppl. Fig.9): if I get it right from Suppl. Fig.9, Leu83 is at the turning point of one helix into another one. Substituting L for P at such a position could evoke major structural changes due to the fixed conformation of proline. Thus, this feature could be much more important (and detrimental for the enzyme activity) than theoretically not changing the hydrophobicity as suggested by the authors. By the way, it is easy to exchange the amino acids and do a modeling of the UGT structure based on published UGT structure. Then L83P could be checked for major alterations.

REVIEWERS' COMMENTS:

Reviewer #1 (Remarks to the Author):

The authors have largely addressed my concerns. I am now able to understand how the GWA was conducted and what the results are indicating.

Reviewer #2 (Remarks to the Author):

From my point of view, the responses are satisfying.

There is only one minor remark to be considered in the discussion (line 389 with respect to Suppl. Fig.9): if I get it right from Suppl.Fig.9, Leu83 is at the turning point of one helix into another one. Substituting L for P at such a position could evoke major structural changes due to the fixed conformation of proline. Thus, this feature could be much more important (and detrimental for the enzyme activity) than theoretically not changing the hydrophobicity as suggested by the authors. By the way, it is easy to exchange the amino acids and do a modeling of the UGT structure based on published UGT structure. Then L83P could be checked for major alterations.

Response: Here we did a modeling of the UGT706D1 structure based on the published UGT structure (PDB: 2acv) by online service (<https://swissmodel.expasy.org/>). We compared the modeling protein structures using either Pro83 or Leu83 (pink and light cyan, respectively) and showed that, in our predicted UGT structure, Pro83 (or Leu83) is at the turning point of a loop into a helix, rather than one helix into another helix (**Fig. a** in below). This also holds true for models generated in Supplementary Fig.9. In addition, no structural difference was observed by substituting of L for P at this position (**Fig. b** in below), which is consistent with the observation at the same position in Supplementary Fig.9. Since Supplementary Fig. 9 gives clear view of major information of the key structure of the

protein and editorial team suggests model analysis is not necessary in this case, we tend not to replace Supplementary Fig. 9 with these pictures. It seems from our modeling that L83P is not crucial in determining the enzyme activity. However, it remains to be tested further. We have included this notion into the revised discussion (lines 391-395).

(a) Alignment of UGT706D1 structures using either Pro83 or Leu83 (pink and light cyan, respectively). Red arrow indicates the mutation L83P cause no structural change at this position (shown in blue). Both loop and helix are marked in red line region.

(b) Close-up view of cartoon diagram showing overlapping structure including amino acid residues 72-97. No difference was observed in blue region (L83P mutation).